# Rapid and stable mobilization of CD8+ T cells by SARS-CoV-2 mRNA vaccine

Valerie Oberhardt[1,2,12], Hendrik Luxenburger[1,3,12], Janine Kemming[1,2,12], Isabel Schulien[1,12], Kevin Ciminski[4], Sebastian Giese[4], Benedikt Csernalabics[1], Julia Lang-Meli[1,3], Iga Janowska[5], Julian Staniek[2,5], Katharina Wild[1,6], Kristi Basho[1], Mircea Stefan Marinescu[1], Jonas Fuchs[4], Fernando Topfstedt[5], Ales Janda[7], Oezlem Sogukpinar[1], Hanna Hilger[1], Katarina Stete[1], Florian Emmerich[8], Bertram Bengsch[1,9], Cornelius F. Waller[10], Siegbert Rieg[1], Sagar[1], Tobias Boettler[1,11], Katharina Zoldan[1], Georg Kochs[4], Martin Schwemmle[4], Marta Rizzi[5], Robert Thimme[1,13 ✉], Christoph Neumann-Haefelin[1,13 ✉] & Maike Hofmann[1,13 ✉]

SARS-CoV-2 spike mRNA vaccines[1–3] mediate protection from severe disease as early as ten days after prime vaccination[3], when neutralizing antibodies are hardly detectable[4–6]. Vaccine-induced CD8+ T cells may therefore be the main mediators of protection at this early stage[7,8]. The details of their induction, comparison to natural infection, and association with other arms of vaccine-induced immunity remain, however, incompletely understood. Here we show on a single-epitope level that a stable and fully functional CD8+ T cell response is vigorously mobilized one week after prime vaccination with bnt162b2, when circulating CD4+ T cells and neutralizing antibodies are still weakly detectable. Boost vaccination induced a robust expansion that generated highly differentiated effector CD8+ T cells; however, neither the functional capacity nor the memory precursor T cell pool was affected. Compared with natural infection, vaccine-induced early memory T cells exhibited similar functional capacities but a different subset distribution. Our results indicate that CD8+ T cells are important effector cells, are expanded in the early protection window after prime vaccination, precede maturation of other effector arms of vaccine-induced immunity and are stably maintained after boost vaccination.

The current SARS-CoV-2 vaccination campaign provides the unique opportunity to gain important insights into human CD8+ T cell biology in the context of prime or boost mRNA vaccination. Initial data revealed that all arms of adaptive immunity such as neutralizing antibodies, virus-specific CD4+ T cells with T helper 1 (T_H1) polarization and IFNγ-producing CD8+ T cells emerge after prime or boost vaccination[4,5,9]. The onset of mRNA vaccine-mediated protection has been observed as early as 10–12 days after the first dose[3]. During this early phase, T cells and spike-specific antibodies are detectable[7,8], whereas neutralizing antibodies first appear after boost[4–6,10,11]. These observations point towards a key role of vaccine-induced T cells in early protection after prime vaccination. Previous studies focused on the analysis of the overall vaccine-elicited spike-reactive T cell response[4,5,7,8,12]; however, by this approach, the strength, dynamics and functional capacity are underestimated or even blurred in contrast to analyses performed at the single epitope level[5]. Here, we conducted continuous longitudinal analyses starting at baseline of prime vaccination until 3–4 months after boost on a single epitope level, to track the trajectories of bnt162b2 vaccine-elicited spike-specific CD8+ T cell responses in comparison to spike-specific CD4+ T cells, B cells, antibodies and their neutralizing activity.

## Vaccine-elicited CD8+ T cells

We longitudinally collected peripheral blood mononuclear cells (PBMCs) and sera in 3–4-day intervals from 32 healthcare workers (Supplementary Table 1) that had not been previously infected with SARS-CoV-2, starting before prime until day 80–120 after boost (Extended Data Fig. 1a) and analysed the induction of spike-specific CD8+ T cells that target A*01/S_865, A*02/S_269 and A*03/S_378 epitopes in 4–5 individuals each (Extended Data Fig. 1b). All three epitopes are not highly conserved between SARS-CoV-2 and SARS-CoV-1, MERS or

[1]Department of Medicine II (Gastroenterology, Hepatology, Endocrinology and Infectious Diseases), Freiburg University Medical Center, Faculty of Medicine, University of Freiburg, Freiburg, Germany. [2]Faculty of Biology, University of Freiburg, Freiburg, Germany. [3]IMM-PACT, Faculty of Medicine, University of Freiburg, Freiburg, Germany. [4]Institute of Virology, Freiburg University Medical Center, Faculty of Medicine, University of Freiburg, Freiburg, Germany. [5]Department of Rheumatology and Clinical Immunology, Freiburg University Medical Center, Faculty of Medicine, University of Freiburg, Freiburg, Germany. [6]Faculty of Chemistry and Pharmacy, University of Freiburg, Freiburg, Germany. [7]Department of Pediatrics and Adolescent Medicine, Ulm University Medical Center, Ulm, Germany. [8]Institute for Transfusion Medicine and Gene Therapy, Freiburg University Medical Center, Faculty of Medicine, University of Freiburg, Freiburg, Germany. [9]Signalling Research Centres BIOSS and CIBSS, University of Freiburg, Freiburg, Germany. [10]Department of Haematology, Oncology & Stem Cell Transplantation, Freiburg University Medical Center, Faculty of Medicine, University of Freiburg, Freiburg, Germany. [11]Berta-Ottenstein Programme, Faculty of Medicine, University of Freiburg, Freiburg, Germany. [12]These authors contributed equally: Valerie Oberhardt, Hendrik Luxenburger, Janine Kemming, Isabel Schulien. [13]These authors jointly supervised this work: Robert Thimme, Christoph Neumann-Haefelin, Maike Hofmann. ✉e-mail: robert.thimme@uniklinik-freiburg.de; christoph.neumann-haefelin@uniklinik-freiburg.de; maike.hofmann@uniklinik-freiburg.de

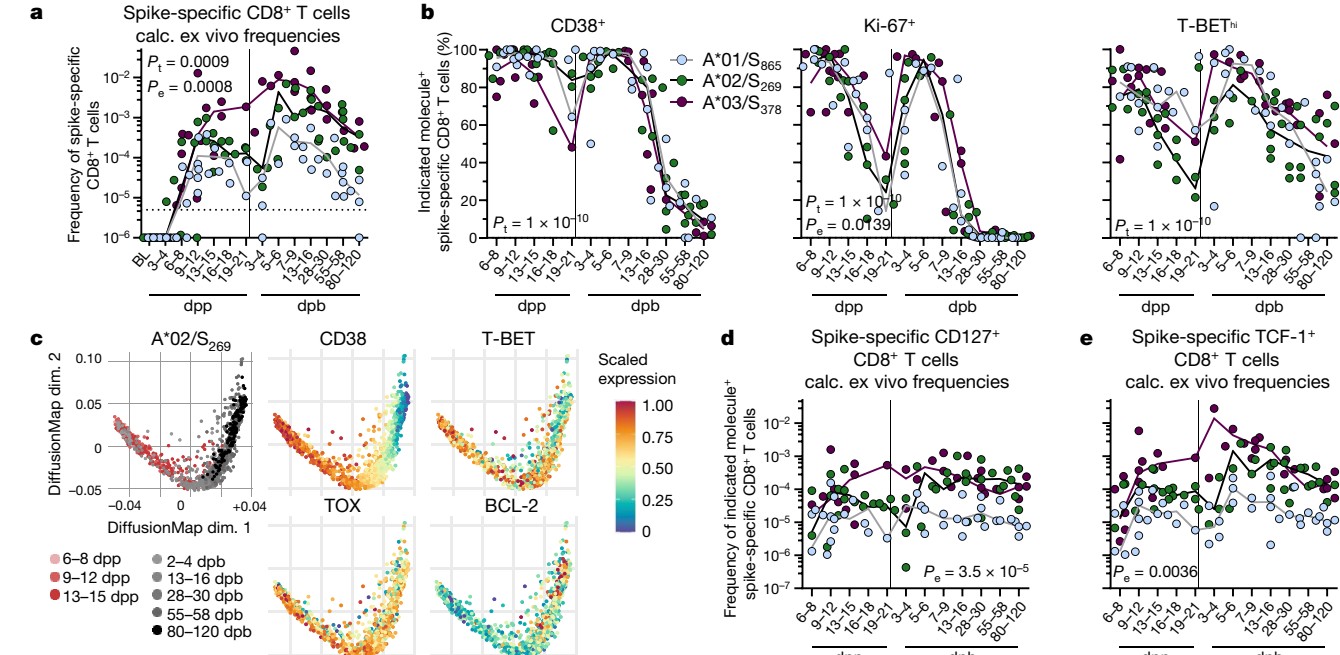

**Fig. 1 | Vaccine-elicited epitope-specific CD8+ T cells. a**, Calculated ex vivo frequency indicated at baseline (BL), dpp and dpb for spike-specific CD8+ T cells. Detection limit: $5 \times 10^{-6}$. **b**, Percentage of CD38, Ki-67 and T-BET[hi] expressing spike-specific non-naive CD8+ T cells. **c**, Diffusion map showing flow cytometry data for A*02/S$_{269}$-specific CD8+ T cells in relation to dpp (shades of red) and dpb (shades of grey) in one individual. Expression levels of

CD38, T-BET, TOX and BCL-2 are plotted on the diffusion map (blue denotes low expression; red denotes high expression). **d**, **e**, Calculated ex vivo frequencies of non-naive spike-specific CD8+ T cells expressing CD127 or TCF-1 for spike-specific CD8+ T cells. Line indicates median. $P$ values determined by two-way ANOVA with main effects only comparing the effect of the different epitopes ($P_e$) and of time course ($P_t$).

common cold coronaviruses (Extended Data Fig. 1c). Thus, the detected spike-specific CD8+ T cells indeed reflect a response to vaccination. The epitopes are not affected by the sequence variations present in the variants of concern (VOC) alpha, beta, gamma and delta (Extended Data Fig. 1c). The tested A*01-, A*02- and A*03-restricted CD8+ T cells that are part of a broader spike-specific CD8+ T cell response, however, proved to be dominant when analysing responses that span the whole S protein (Extended Data Fig. 1d). Ex vivo frequencies of A*01/S$_{865}$-, A*02/S$_{269}$- and A*03/S$_{378}$-specific CD8+ T cells were rather low after vaccination (Extended Data Fig. 2a). To increase the detection rate and to allow subsequent comprehensive profiling, we performed pMHCI-tetramer enrichment (Extended Data Fig. 2b). We detected a rapid and substantial induction of spike-specific CD8+ T cells that were present in 9 out of 13 tested donors already at days 6–8 and peaked in most donors 9–12 days post prime (dpp) (Fig. 1a). The strong CD8+ T cell activation was also reflected by high expression of CD38 and Ki-67 as early as days 6–8 in most cells (Fig. 1b, c and Extended Data Fig. 2c). Boost vaccination led to a further increase of CD8+ T cell frequencies that peaked 5–6 days post boost (dpb) with a subsequent slow contraction phase that reached nearly pre-boost frequencies at about 80–120 dpb (Fig. 1a). Post-boost and post-prime expansion were accompanied by effector T (T$_{eff}$) cell differentiation (high expression of Ki-67, CD38, granzyme B, PD-1, CD39, T-BET and TOX) (Fig. 1b and Extended Data Fig. 2c–e). However, $t$-distributed stochastic neighbour embedding ($t$-SNE) analysis revealed that CD8+ T$_{eff}$ cells are qualitatively different at the peak expansion after boost (obtained at 5–6 dpb) compared with prime (obtained at 9–12 dpp) with a more consolidated cytotoxic effector cell phenotype (increased expression of T-BET, TOX and CD39) post boost (Extended Data Fig. 3a). This consolidated post-boost T$_{eff}$ cell response is further supported by diffusion map analysis (Fig. 1c and Extended Data Fig. 3b). Specifically, diffusion map embedding revealed a continuous relationship of the longitudinally collected spike-specific CD8+ T cells after

prime (depicted in reddish colours)/boost (depicted in grey colours) indicating a directed trajectory of the T$_{eff}$ cell response. Along the trajectory, CD8+ T cells exhibited the highest expression of PD-1, TOX, T-BET and CD38 after boost indicating profound activation and progressing differentiation (Fig. 1c and Extended Data Fig. 3b). Of note, a single vaccine dose also induced boost expansion and strong activation but lower TOX expression (Extended Data Fig. 4a–c) of spike-specific CD8+ T cells in individuals who recovered from mild to moderate infection approximately 12 months before vaccination (Supplementary Table 1).

We also assessed the induction of spike-specific memory precursor CD8+ T cells that are characterized by CD127, BCL-2 and TCF-1 expression and are relevant for maintaining the CD8+ T cell response[13,14]. Roughly 20–30% of spike-specific CD8+ T cells expressed CD127 after prime followed by a transient reduction and subsequent strong increase after boost (Fig. 1d and Extended Data Fig. 4d). Expression dynamics of TCF-1 (Fig. 1e and Extended Data Fig. 4e) and BCL-2 (Extended Data Fig. 4f) were similar to CD127. However, the overall frequency of CD127+ (Fig. 1d) and TCF-1+ (Fig. 1e) spike-specific CD8+ T cells remained constant indicating a stable memory precursor pool induced already early after prime vaccination. Together, bnt162b2 vaccination vigorously induces a lasting spike-specific CD8+ T cell response rapidly after prime vaccination.

## CD8+ T cell function after vaccination

After two weeks of peptide-specific in vitro expansion (Extended Data Fig. 5a, b), we detected higher frequencies of spike-specific CD8+ T cells after boost compared to prime vaccination (Extended Data Fig. 5c, d). However, the expansion index, a measure taking the input number of virus-specific CD8+ T cells into account was comparable for spike-specific CD8+ T cells after prime and boost vaccination, but differed between the A*01/S$_{865}$- A*02/S$_{269}$- and A*03/S$_{378}$-specific CD8+ T cell

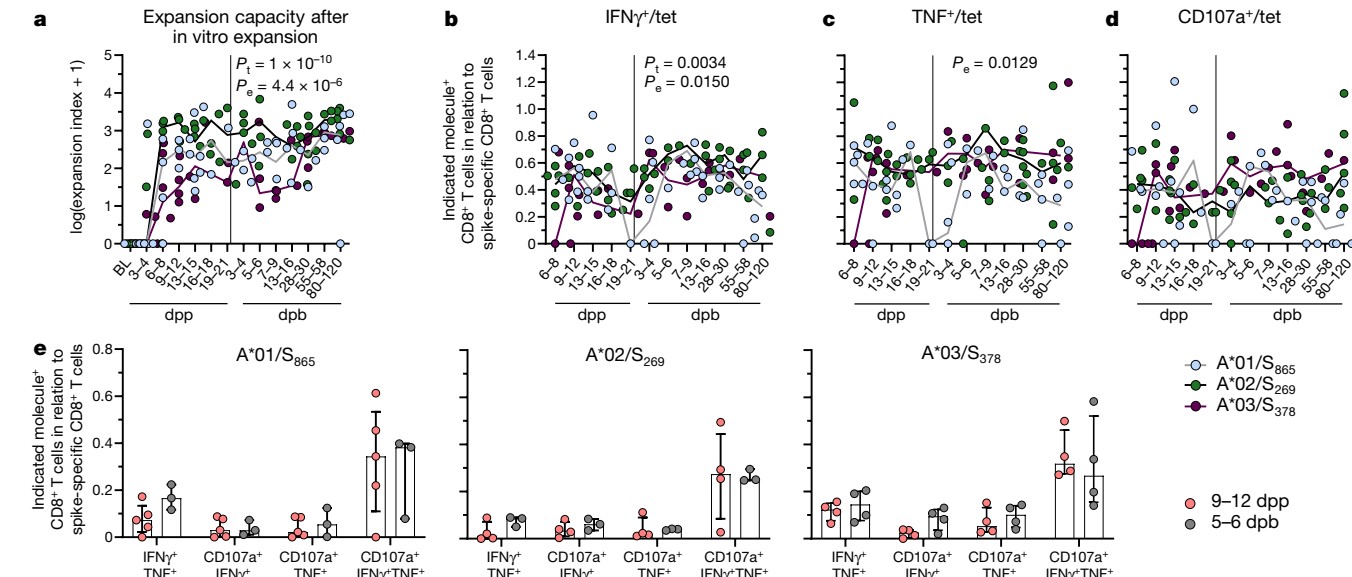

**Fig. 2 | Functional capacities of vaccine-elicited spike-specific CD8[+] T cells.**
**a**, Expansion capacity of spike-specific CD8[+] T cells after in vitro expansion.
**b**–**d**, Percentage of CD8[+] T cells producing effector molecules related to
the frequency of spike-specific CD8[+] T cells. **e**, Bar graphs depicting the
polyfunctionality of spike-specific CD8[+] T cells comparing 9–12 dpp and

5–6 dpb vaccination. Line indicates median. Bar charts show the median with
interquartile range (IQR). *P* values determined by two-way ANOVA with main
effects only comparing the effect of the different epitopes and of time course
(**a**–**d**) or by Mann–Whitney test with Holm–Šídák method (**e**).

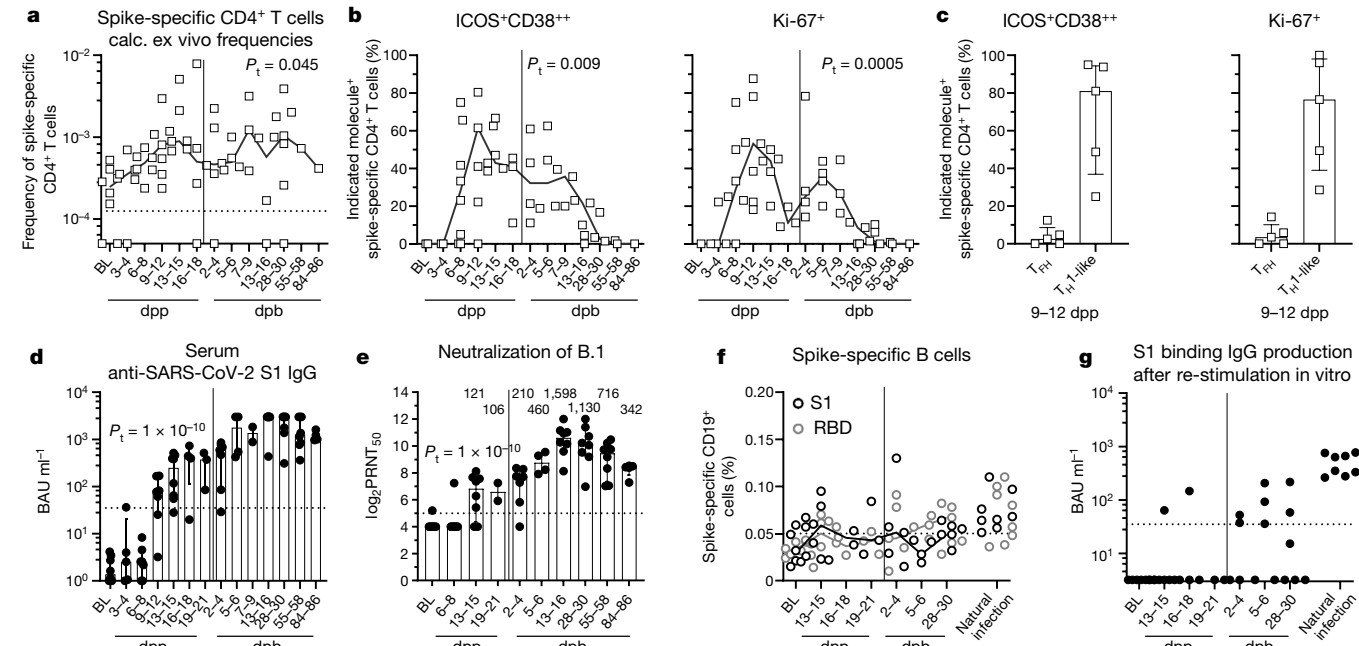

**Fig. 3 | Circulating spike-specific CD4[+] T cells, B cells and antibodies.**
**a**, Calculated ex vivo frequency of DRB1*15:01/S$_{236}$-specific CD4[+] T cells ex vivo
after pMHCII tetramer-based enrichment is indicated at baseline, dpp and dpb.
Detection limit: $1.25 \times 10^4$. **b**, ICOS[+]CD38[++] and Ki-67 expression within
non-naive, DRB1*15:01/S$_{236}$-specific CD4[+] T cells. **c**, ICOS[+]CD38[++] and
Ki-67-expressing non-naive DRB1*15:01/S$_{236}$-specific CD4[+] T cells on 9–12 dpp
within T$_{FH}$ (CXCR5[+]PD-1[+]) and T$_H$1-like (CXCR5[-]CXCR3[+]) cells. **d**, Anti-SARS-CoV-2
spike IgG at baseline and after vaccination (<35.2 binding antibody units (BAU)
per ml: negative, ≥35.2 BAU ml[−1]: positive; upper limit of quantification:
3,000 BAU ml[−1]). **e**, Antibody neutralization activity is depicted as 50% plaque
reduction neutralization tests (PRNT$_{50}$) at baseline, dpp and dpb vaccination

for the SARS-CoV-2 variant B.1. Numbers indicate non-logarithmic median
value. Detection limit: 5 log$_2$PRNT$_{50}$. **f**, Percentage spike-specific B cells
depicted at baseline, dpp and dpb as well as in natural infection for S1 and RBD.
Detection limit: 0.05%. **g**, Secreted anti-SARS-CoV-2 spike IgG from PBMCs after
in vitro stimulation with CpG and IL-2 (<35.2 BAU ml[−1]: negative, ≥35.2 BAU ml[−1]:
positive). Line indicates median. Bar charts show the median and IQR. *P* values
determined by one-way ANOVA with a mixed effects model comparing the
effect of the time course (**a**, **b**, **f**), a Wilcoxon test (**c**) or a two-way ANOVA with
main effects only comparing the effect of the different epitopes and of time
course (**g**).

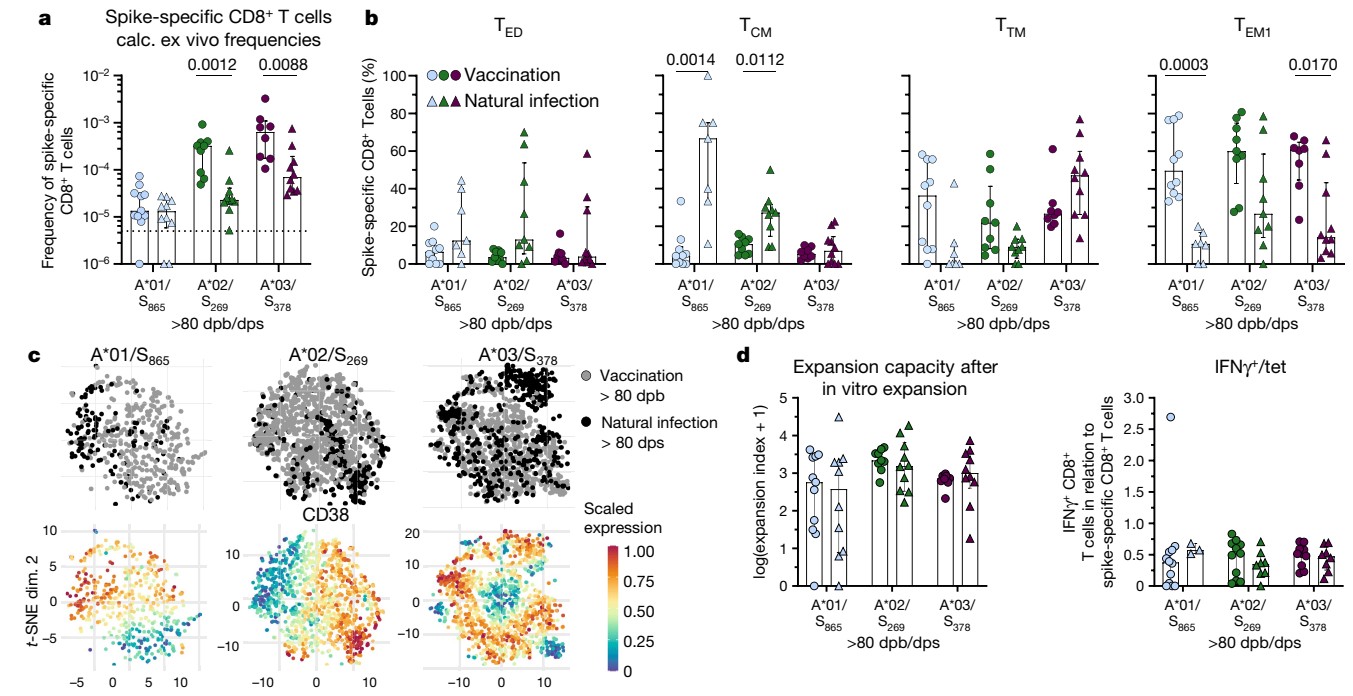

**Fig. 4 | Early memory CD8⁺ T cells after vaccination and natural infection.** **a**, Calculated frequency of spike-specific CD8⁺ T cells 80–200 dpb vaccination or dps in natural infection. Detection limit: $5 \times 10^{-6}$. **b**, Distribution of spike-specific CD8⁺ T cell memory subsets 80–200 dpb/dps. **c**, t-SNE representation of flow cytometry data, depicting spike-specific CD8⁺ T cells more than 80 dpb vaccination and dps of natural infection (grey: vaccination, black: natural infection) for A*01/S₈₆₅- (vaccination n = 9, natural infection n = 9),

A*02/S₂₆₉-(vaccination n = 10, natural infection n = 8) and A*03/S₃₇₈- (vaccination n = 9, natural infection n = 9) specific CD8⁺ T cells. **d**, Left, expansion index of spike-specific CD8⁺ T cells after in vitro expansion at 80-200 dpb/dps. Right, percentage of IFNγ-producing CD8⁺ T cells related to the frequency of spike-specific CD8⁺ T cells after in vitro expansion at 80–200 dpb/dps. T_TM, transitional memory T cells. Bar charts show the median with IQR. P values were determined by Mann–Whitney test with Holm–Šídák method.

responses (Fig. 2a). Thus, the increased frequencies of spike-specific CD8⁺ T cells after peptide-specific expansion most probably result from the increased ex vivo frequencies after boost. We also assessed spike-specific production of IFNγ and TNF (Extended Data Fig. 5e, f) and degranulation as indicated by CD107a expression (Extended Data Fig. 5g) in relation to the frequency of spike-specific CD8⁺ T cells after expansion as a measure of the effector function per cell. We observed reasonable effector capacity of circulating spike-specific CD8+ T cells obtained as early as 6–8 dpp (Fig. 2b–d). Similar to the expansion capacity, cytokine production and degranulation capacity remained nearly stable after boost compared to prime (Fig. 2b–e). Hence, functionally competent spike-specific CD8⁺ T cells that target different epitopes are substantially induced early after prime, and subsequent boost vaccination does not further increase their functional capacities in vitro.

## CD4⁺ T cells, B cells and antibodies

Next, we longitudinally assessed circulating spike-specific CD4⁺ T cells that target DRB1*15:01/S₂₃₆ (Extended Data Fig. 6a) after prime and boost vaccination in eight individuals (Supplementary Table 1). The selected DRB1*15:01/S₂₃₆ epitope is unique for SARS-CoV-2 in comparison to SARS-CoV-1, MERS or common cold coronaviruses and conserved in circulating SARS-CoV-2 variants (B.1, alpha, gamma and delta) except for VOC beta (Extended Data Fig. 6c). The frequencies of DRB1*15:01/S₂₃₆-specific CD4⁺ T cells were lower than CD8⁺ T cell responses but detectable after pMHCII tetramer-based enrichment (Extended Data Fig. 6b). At baseline and in historic control samples (banked before August 2019), spike-specific CD4⁺ T cells were detectable with a primarily naive phenotype (Extended Data Fig. 6d, e), which reflects the presence of antigen-unexperienced precursors. After vaccination, the proportion of naive spike-specific CD4⁺ T cells decreased, which

suggests vaccine-induced activation (Extended Data Fig. 6e). However, compared with CD8⁺ T cells, we observed a lower mobilization of circulating spike-specific CD4⁺ T cells indicated by a limited increase of frequencies (Fig. 3a) and a smaller percentage of activated ICOS⁺CD38⁺⁺ or Ki-67⁺ subsets (Fig. 3b and Extended Data Fig. 6f). Most activated DRB1*15:01/S₂₃₆-specific CD4⁺ T cells exhibited a T_H1 cell phenotype (Fig. 3c). In line with this observation, vaccine-induced spike-specific CD4⁺ T cells displayed a T_H1 cell rather than a follicular helper T (T_FH) cell phenotype (Extended Data Fig. 6g).

We then assessed the kinetics of the vaccine-induced humoral response. The distribution of peripheral B cell subpopulations was stable throughout prime or boost vaccination, with the exception of a progressively slight increase in antibody-secreting cells (ASC) (Extended Data Fig. 7a, b). An increase in the frequency of CD95⁺ B cells was observed shortly after boost, which indicates ongoing B cell activation via CD40-mediated T cell help and/or B cell receptor activation within secondary lymphoid organs[15] (Extended Data Fig. 7b). In line with the appearance of activated B cells in the periphery, we observed a progressive maturation of the serum antibody response with S1-specific IgM present after prime whereas S1-specific IgG reasonably detectable after boost (Fig. 3d and Extended Data Fig. 7c), coinciding with a high neutralization capacity in SARS-CoV-2 plaque reduction assays. More precisely, SARS-CoV-2 B.1 and VOC alpha were similarly well neutralized by post-boost sera, whereas the cross-neutralization activity against VOC beta was reduced approximately by a factor of 5 (Fig. 3e and Extended Data Fig. 7d). Neutralization capacity of post-boost sera was clearly increased compared with time point-matched mild infection (Extended Data Fig. 7e). In line with the progressive maturation of the antibody response, S1- and receptor-binding domain (RBD)-specific B cells (Extended Data Fig. 7f) largely remained below the ex vivo detection limit until the first week post boost (Fig. 3f). The

delayed appearance of circulating S1-specific B cells was confirmed by polyclonal restimulation in vitro (Fig. 3g), which showed a limited presence of class-switched B cells that could produce S1-specific IgG before boost. S1-specific B cells were largely unswitched after prime (Extended Data Fig. 7g, h), also reflected by S1-specific IgM production upon polyclonal restimulation in vitro (Extended Data Fig. 7c), and acquired a memory phenotype after boost vaccination (Extended Data Fig. 7g, h). In addition, after boost vaccination, S1-specific B cells showed increased transferrin receptor (CD71) and CD95 expression (Extended Data Fig. 7g, h), which indicates their germinal centre origin[16]. Hence, bnt162b2 vaccination efficiently elicits a protective humoral immune response, composed of ASC and antigen-specific memory B cells that are mobilized to the periphery after boost.

## Early memory CD8+ T cells

We compared vaccine-elicited spike-specific early memory CD8+ T cells (days post boost vaccination) with time point-matched T cells induced by natural infection (days post symptom onset) (Extended Data Fig. 8a). A*01/$S_{865}$-specific CD8+ T cell frequencies were similar after vaccination versus infection at all time points analysed. However, in comparison to vaccination, lower frequencies of A*02/$S_{269}$- and A*03/$S_{378}$-specific CD8+ T cells were detectable at days 80–120 (6 out of 30 (natural infected), 4 out of 28 (vaccinees) were obtained at days 120–200) after natural infection (Fig. 4a and Extended Data Fig. 8b). Phenotypic characteristics of early memory CD8+ T cells targeting A*01/$S_{865}$, A*02/$S_{269}$ and A*03/$S_{378}$ differed after vaccination versus natural infection as revealed by t-SNE analyses (Extended Data Fig. 8c). Possible reasons for this include differences in their MHCI binding and presentation characteristics (Extended Data Fig. 1c). In addition, we also observed differences in T cell memory subset distribution (Fig. 4b and Extended Data Fig. 9a–c) of spike-specific early memory CD8+ T cells with higher fractions of more early differentiated subsets, for example, early differentiated ($T_{ED}$) and central memory ($T_{CM}$) T cells for A*01/$S_{865}$- and A*02/$S_{269}$-specific CD8+ T cells and transitional memory cells for A*03/$S_{378}$-specific CD8+ T cells after natural infection (80–120 dps). By contrast, higher frequencies of effector memory 1 T cells ($T_{EM1}$) were detectable after vaccination (80–120 dpb) (Fig. 4b). Spike-specific effector memory 2 and 3 T cells ($T_{EM2}$ and $T_{EM3}$) and terminally differentiated effector memory T cells that expressed CD45RA ($T_{EMRA}$) were hardly detectable in the circulation (Extended Data Fig. 9c). Of note, the memory subset distribution of A*03/$S_{378}$-specific CD8+ T cells differed from A*01/$S_{865}$- and A*02/$S_{269}$-specific CD8+ T cells with only a minor fraction of $T_{ED}$ and $T_{CM}$ cells targeting A*03/$S_{378}$ reflecting an overall further differentiation towards effector memory subsets (Fig. 4b and Extended Data Fig. 9b). t-SNE analysis of concatenated expression data further supports qualitative differences of spike-specific CD8+ T cells obtained from the early memory phase (80–120 dpb/dps) after vaccination compared to natural infection being less pronounced for A*03/$S_{378}$-specific CD8+ T cells (Fig. 4c). For A*01/$S_{865}$-specific CD8+ T cells we also observed higher expression of TCF-1 and BCL-2 after natural infection (Extended Data Fig. 10a, b). Both t-SNE analysis and manual gating demonstrated a higher and prolonged CD38 expression on spike-specific CD8+ T cells after natural infection (Fig. 4c and Extended Data Fig. 10c). However, vaccine- and natural infection-associated expansion capacity and cytokine production of spike-specific CD8+ T cells were similar (Fig. 4d and Extended Data Fig. 10d, e). Hence, compared with natural infection, vaccine-associated spike-specific early memory CD8+ T cell populations exhibit similar functional capacities but a different subset distribution.

## Discussion

In summary, a robust, stable and fully functional spike-specific CD8+ T cell response is elicited already after prime vaccination at a time point when neutralizing antibodies were hardly detectable and coincides with the protective effect observed for mRNA vaccines that starts at 10–12 dpp[2,3]. In contrast to CD8+ T cells, peak mobilization of neutralizing antibodies and antigen-specific B cells to the periphery was first detectable after boost. This is in line with previous reports[4,7,11,12] and most probably represents maturation of the response in secondary lymphoid organs[17] with subsequent release to the circulation. After boost, highly cross-neutralizing antibodies are present in the sera, clearly adding a major protective effector mechanism on top of the early-mobilized spike-specific CD8+ T cell response. The humoral and CD8+ T cell response are potentially coordinated by early elicited spike-reactive CD4+ T cells[8] that underwent a limited boost expansion after second dose mRNA vaccination supporting their coordinating role.

Fully functional vaccine-elicited early memory CD8+ T cells patrol the periphery for SARS-CoV-2 at least within the first months. The functional capacity of spike-specific early memory CD8+ T cells is similar after vaccination and natural infection up to 3–4 months after boost or symptom onset. Compared with natural infection, however, the early memory pool of spike-specific CD8+ T cells after vaccination exhibits a different memory T cell subset distribution that may affect long-term maintenance characteristics[18]. This difference may be caused by differential duration and location of antigen contact and different inflammatory responses after vaccination versus infection[19,20], as indicated by a lower CD38 expression on early memory spike-specific CD8+ T cells after vaccination compared with natural infection[4,21]. Follow-up studies including larger cohorts of vaccinees and SARS-CoV-2 convalescent individuals are clearly required to assess longevity of CD8+ T cell immunity. Our study was limited to circulating spike-specific adaptive immunity, and did not address local immunity at the viral entry site, the respiratory tract. However, our data provide insights into the protective mechanisms that underlie bnt162b2 vaccination with implications for the development of vaccination strategies against emerging pathogens and cancer.

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

## Methods

### Study cohort

In total, 32 healthcare workers that received a prime and boost vaccination with the mRNA vaccine bnt162b2/Comirnaty, 59 convalescent individuals following a mild course of SARS-CoV-2 infection, 2 convalescent individuals given one dose of bnt162b2/Comirnaty 12 months after infection, and historic controls (sampled before August 2019) of 8 healthy individuals were recruited at the Freiburg University Medical Center, Germany. A mild course of infection was characterized by clinical symptoms without respiratory insufficiency. SARS-CoV-2 infection was confirmed by positive PCR testing from oropharyngeal swab and/or SARS-CoV-2 spike IgG positive antibody testing. Donor characteristics are summarized in Supplementary Table 1. HLA-typing was performed by next-generation sequencing and is listed in Supplementary Table 1.

### Ethics

Written informed consent was obtained from all participants and the study was conducted according to federal guidelines, local ethics committee regulations (Albert-Ludwigs-Universität, Freiburg, Germany; vote: 322/20, 21-1135 and 315/20) and the Declaration of Helsinki (1975).

### PBMC isolation

Venous blood samples were collected in EDTA-anticoagulated tubes. PBMCs were isolated with lymphocyte separation medium density gradients (Pancoll separation medium, PAN Biotech GmbH) and stored at −80 °C. Frozen PBMCs were thawed in complete medium (RPMI 1640 supplemented with 10% fetal calf serum, 1% penicillin/streptomycin and 1.5% HEPES buffer 1 M (all additives from Thermo Scientific) containing 50 U ml$^{-1}$ benzonase (Sigma).

### Sequence alignment

Sequence homology analyses were performed in Geneious Prime 2020.0.3 (https://www.geneious.com/) using Clustal Omega 1.2.2 alignment with default settings[22]. Reference genomes of human coronaviruses 229E (NC_002645), HKU1 (NC_006577), NL63 (NC_005831), OC43 (NC_006213), MERS (NC_019843), SARS-CoV-1 (NC_004718) and SARS-CoV-2 (MN908947.3) were downloaded from NCBI database. Spike proteins of human coronaviruses were aligned according to their homology (amino acid level). Analysed spike SARS-CoV-2 epitopes were then mapped to the corresponding protein alignment. Correspondingly mutation analyses were performed with the spike protein of VOC alpha, beta, gamma and delta.

### In vitro expansion and intracellular IFNγ staining with overlapping peptides

A total of 182 overlapping peptides that spanned the SARS-CoV-2 spike sequence (Gene Bank Accession code MN908947.3) were synthesized as 18-mers overlapping by 11 amino acids with a free amine NH$_2$ terminus and a free acid COOH terminus with standard Fmoc chemistry and a purity of >70% (Genaxxon Bioscience). In vitro expansion with OLPs was performed as follows: 20% of the PBMCs were stimulated with a pool of all 181 SARS-CoV-2 spike OLPs (10 µg ml$^{-1}$) for 1 h at 37 °C, washed and co-cultured with the remaining PBMCs in RPMI medium supplemented 20 U ml$^{-1}$ with recombinant IL-2. On day 10, intracellular IFNγ staining was performed with pooled OLPs (45 pools with 4 OLP each). Therefore, cells were re-stimulated with OLP pools (50 µM), DMSO as negative control or PMA and ionomycin as positive control in the presence of brefeldin A an IL-2. After 5 h of incubation at 37 °C, cells were stained for surface markers (CD8$^+$, CD4$^+$; Viaprobe) and intracellular markers (IFNγ). Subsequently, on day 12 the single overlapping peptides of positive pools were tested by intracellular cytokine staining. Viral amino acid sequences of positive individual OLPs were analysed for pre-described minimal epitopes or the best HLA-matched predicted candidate using the Immune Epitope Database website (using two prediction algorithms ANN 4.0 and NetMHCpan EL 4.1[23] for 8-mer, 9-mer and 10-mer peptides with half-maximal inhibitory concentration (IC$_{50}$) of <500 nM).

### Peptides and tetramers for T cell analysis

Peptides were synthesized with an unmodified N terminus and an amidated C terminus with standard Fmoc chemistry and a purity of >70% (Genaxxon Bioscience). Peptide was loaded on HLA class I easYmers (immunAware) according to manufacturer's instructions (A*01/S$_{865}$ LTDEMIAQY, A*02/S$_{269}$ YLQPRTFLL and A*03/S$_{378}$ KCYGVSPTK). SARS-CoV-2 peptide-loaded HLA class I tetramers were produced by conjugation of biotinylated peptide-loaded HLA class I easYmers with phycoerythrin (PE)-conjugated streptavidin (Agilent) according to the manufacturer's instructions. A SARS-CoV-2-specific HLA class II custom tetramer (DRB1*15:01/S$_{236}$ TRFQTLLALHRSYLT) was obtained from (MBL).

### In vitro expansion of spike-specific CD8$^+$ T cells and assessment of effector function

Approximately $1.5 × 10^6$ PBMCs were stimulated with A*01/S$_{865}$, A*02/S$_{269}$ or A*03/S$_{378}$-specific peptides (5 µM) and anti-CD28 monoclonal antibody (0.5 µg ml$^{-1}$, BD) and expanded for 14 days in complete RPMI culture medium containing rIL-2 (20 IU ml$^{-1}$, StemCell Technologies). Intracellular cytokine production and degranulation was assessed with spike-specific peptides (15 µM) in the presence of anti-CD107a (H4A3, 1:100) (BD Bioscience) for 1 h at 37 °C. Afterwards, brefeldin A (GolgiPlug, 0.5 µl ml$^{-1}$) and monensin (GolgiStop, 0.5 µl ml$^{-1}$) (all BD Biosciences) were added for additional 5 h, followed by surface and intracellular staining. The expansion capacity was calculated based on peptide-loaded HLA class I tetramer staining as previously described[24].

### Magnetic bead-based enrichment of spike-specific CD8$^+$ T cells

Spike-specific CD8$^+$ T cells were enriched as previously described[25]. In brief, $1 × 10^7$–$2 × 10^7$ PBMCs (with an average of 15.7% CD8$^+$ T cells) were labelled with PE-coupled peptide-loaded HLA class I tetramers for 30 min. Enrichment was then performed using anti-PE beads with MACS technology (Miltenyi Biotec) according to the manufacturer's instructions. Subsequently, enriched spike-specific CD8$^+$ T cells were analysed by multiparametric flow cytometry and frequencies of spike-specific CD8$^+$ T cells were calculated as described before[25]. Only enriched samples with ≥5 spike-specific CD8 T cells were included in further analyses, resulting in a detection limit of $5 × 10^{-6}$.

### Magnetic bead-based enrichment of spike-specific CD4$^+$ T cells

Enrichment of spike-specific CD4$^+$ T cells was adapted from the method described previously[25]. In brief, $1.5 × 10^7$–$2 × 10^7$ PBMCs of DRB1*15:01-positive donors were labelled with PE-coupled peptide-loaded MHC class II tetramers for 40 min. Then, 5 µl was taken from 1,000 µl pre-enriched sample (1:200) and used for subsequent flow cytometric staining. Subsequent enrichment was performed with anti-PE beads using MACS technology (Miltenyi Biotec) according to the manufacturer's protocol. Enriched spike-specific CD4$^+$ T cells and the pre-enriched sample were used for flow cytometric staining. The complete pre-enriched and enriched samples were recorded. Only enriched samples with ≥5 spike-specific CD4$^+$ T cells were included in further analyses. The frequency of spike-specific CD4$^+$ T cells was calculated as follows: Absolute number of spike-specific CD4$^+$ T cells (enriched sample) divided by the absolute number of CD4$^+$ T cells (pre-enriched sample) × 200. The detection limit as a frequency was calculated as follows: 5 spike-specific CD4$^+$ T cells (enriched sample) divided by the mean number of CD4$^+$ T cells (pre-enriched sample) throughout all tested donors × 200.

### Multiparametric flow cytometry for T cell analysis

The following antibodies were used for multiparametric flow cytometry: anti-CCR7-PE-CF594 (150503, 1:50), anti-CCR7-BUV395 (3D12,

1:25), anti-CD4-BV786 (L200, 1:200), anti-CD8-BUV395 (RPA-T8, 1:400), anti-CD8-BUV510 (SK1, 1:100), anti-CD8-APC (SK-1, 1:200), anti-CD11a-BV510 (HI111, 1:25), anti-CD28-BV421 (CD28.2, 1:100), anti-CD38-APC-R700 (HIT2, 1:400), anti-CD38-BUV737 (HB7, 1:200), anti-CD39-BV650 (TU66, 33:1), anti-CD45RA-BUV496 (HI100, 1:800), anti-CD45RA-BUV737 (HI100, 1:200), anti-CD69-BUV395 (FN50, 1:50), anti-CD107a-APC (H4A3, 1:100), anti-CD127-BUV737 (HIL-7R-M21, 1:50), anti-CD127-BV421 (HIL-7R-M21, 3:100), anti-EOMES-PerCP-eF710 (WD1928, 1:50), anti-Granzyme B-PE-CF594 (GB11, 1:100), anti-ICOS-BV711 (DX29, 1:100), anti-IFN-γ-FITC (25723.11, 1:8), anti-IL-21-PE (3A3-N2.1, 1:25), anti-PD-1-BV605 (EH12.1, 1:50), anti-PD-1-PE-Cy7 (EH12.2H7, 1:200), anti-PD-1-BV786 (EH12.1, 1013122, 3:100), anti-T-BET-PE-CF594 (O4-46,93533305, 3:100), anti-TNF-PE-Cy7 (Mab11, 1:400) (BD Biosciences), anti-BCL-2-BV421 (100, 1:200), anti-CCR7-BV785 (G043H7, 1:50), anti-CD4-AlexaFluor700 (RPA-T4, 300526, 1:200), anti-CD25-BV650 (BC96, 1:33), anti-CD57-BV605 (QA17A04, 1:100), anti-CD127-BV605 (A019D5, 3:100), anti-CXCR3-PerCP-Cy5.5 (G025H7, 1:33), anti-CXCR3-BV510 (G025H7, 3:100), anti-CXCR5-BV421 (J252D4, 1:100), anti-IL-2-PerCP-Cy5.5 (MQ1-17H12, 1:100), anti-Ki-67-BV711 (Ki-67, 1:200), anti-Ki-67-PE-Cy7 (Ki-67, 1:200) (BioLegend), anti-TCF-1-AlexaFluor488 (C63D9, 1:100) (Cell Signaling), anti-CD14-APC-eFluor780 (61D3, 1:400), anti-CD19-APC-eFluor780 (HIB19, 1:400), anti-CD27-FITC (0323, 1:100), anti-KLRG1-BV711 (13F12F2, 1:50), anti-T-BET-PE-Cy7 (4B10, 1:200), anti-TOX-eFluor660 (TRX10, 1:100) (Thermo Fisher), anti-CD45RA-PerCP-Cy5.5 (HI100, 3:100) (Invitrogen). For live/dead discrimination a fixable Viability Dye (APC-eFluor780 1:200, 1:400) (Thermo Fisher) or ViaProbe (7-AAD, 1:33) (BD Biosciences)) was used. FoxP3/Transcription Factor Staining Buffer Set (Thermo Fisher) and Fixation/Permeabilization Solution Kit (BD Biosciences) were used according to the manufacturer's protocol to stain for intranuclear and cytoplasmic molecules, respectively. After fixation of cells in 2% paraformaldehyde (PFA, Sigma), analyses were performed on FACSCanto II, LSRFortessa with FACSDiva software version 10.6.2 (BD) or CytoFLEX (Beckman Coulter) with CytExpert Software version 2.3.0.84. Data were analysed with FlowJo 10.6.2 (Treestar).

### Dimensional reduction of multiparametric flow cytometry data
Dimensionality reduction of multiparametric flow cytometry data was done with R version 4.0.2 using the Bioconductor (release (3.11)) CATALYST package23. The analyses were performed on gated virus-specific CD8+ T cells including the markers CD69, CD45RA, BCL-2, PD1, CD25, Ki-67, TCF-1, EOMES, CCR7, T-BET, TOX and CD38. Downsampling of cells to 100 or 200 cells ($t$-SNE or diffusion maps) was performed before dimensionality reduction to facilitate the visualization of different samples. Marker intensities were transformed by arcsinh (inverse hyperbolic sine) with a cofactor of 150. Dimensionality reduction on the transformed data was achieved by $t$-SNE and diffusion map visualization.

### S1- and RBD-tetramerization for B cell analysis
A biotinylated form of recombinant S1 and RBD proteins (BioLegend) were tetramerized by addition of PE-conjugated or BV421-conjugated streptavidin (BioLegend) and used for B cell tetramer staining assays. In brief, streptavidin-PE or streptavidin-BV21 was added in an amount that equals one-fifth of the monomer substrate amount. The streptavidin was added in five equal portions to the monomer and incubated each time at 4 °C for 20 min on a shaker. The tetramers were filled up to 100 μl with 0.1% BSA in PBS and stored at 4 °C.

### Multiparametric flow cytometry for B cell analysis
Phenotype of vaccinated individuals' PBMCs was determined by flow cytometry with the following antibodies: anti-CD20-BV510 (2H7, 1:80), anti-IgM-BV605 (MHM-88, 1:200), anti-CD71-FITC (CY1G4, 1:1000), anti-CD95-PE-Dazzle594 (DX2, 1:50), anti-CD24-FITC (ML5, 1:100), anti-CD38-PE-Cy7 (HB-7, 1:300), anti-BAFF-R-AF647 (11C1, 1:100),

anti-CD19-APC-Cy7 (HIB19, 1:150) (BioLegend); anti-IgG-BV650 (G18-145, 1:600), anti-CD27-BV786 (L128, 1:100), anti-CD69-BV480 (FN50, 1:200) (BD Biosciences); anti-IgA-PerCP (polyclonal, 1:200) (Jackson ImmunoResearch); anti-CD3-SB-436 (OKT3, 1:200), anti-CD33-Super Bright 436 (WM-53, 1:50), anti-IgD-PerCP-eFluor 710 (IA6-2, 1:200) (Invitrogen). Dead cell exclusion was performed by Zombie NIR Fixable Viability Kit (Biolegend, 1:800). Multiparametric flow cytometry data was collected on Cytek Aurora with SpectroFlo Software version 2.2.0.3.

### In vitro PBMCs activation and ELISA
PBMCs of vaccinated individuals and patients with a history of SARS-CoV-2 infection were plated at $0.5 \times 10^6$ cells ml$^{-1}$ and polyclonally stimulated for 9 days with thiol-modified CpG (0.25 μM, TCGTCGTTTT-GTCGTTTTGTCGTT) and hIL-2 (100 ng/ml, Immunotools). At day 9, the supernatants of the in vitro culture were cleared from debris by centrifugation and used to determine the presence of SARS-CoV-2 spike-specific IgG antibodies (Anti-SARS-CoV-2-QuantiVac-ELISA (IgG), Euroimmun) according to the manufacturer's instructions. To detect S1 specific IgM, supernatant of the in vitro culture and serum of vaccinated individuals was incubated on a S1 pre-coated plate (Anti-SARS-COV-2, Euroimmun). Bound IgM was detected with alkaline phosphatase-conjugated anti-human IgM (Jackson ImmunoResearch), and developed with p-nitrophenyl phosphate (Sigma-Aldrich) in DEA buffer.

### Serum IgG determination
SARS-CoV-2-specific antibodies were determined by Anti-SARS-CoV-2-QuantiVac-ELISA (IgG) from Euroimmun detecting anti-SARS-CoV-2 spike IgG (anti-SARS-CoV-2 S IgG; <35.2 BAU ml$^{-1}$: negative, ≥ 35.2 BAU ml$^{-1}$: positive) according to the manufacturer's instructions.

### Neutralization assay
Samples of vaccinated and convalescent individuals were tested in a plaque reduction neutralization assay. In brief, VeroE6 cells were seeded in 12-well plates at a density of $2.8 \times 10^5$ cells per well 24 h before infection. Serum samples were diluted at ratios of 1:16, 1:32, 1:64, 1:128, 1:256, 1:512 and 1:1,024 in 50 μl PBS total volume. For each sample, one negative control was included (PBS without serum). Diluted sera and negative controls were subsequently mixed with 90 plaque-forming units (PFU) of authentic SARS-CoV-2 (either B.1, alpha or beta variant) in 50 μl PBS (1,600 PFU ml$^{-1}$) resulting in final sera dilution ratios of 1:32, 1:64, 1:128, 1:256, 1:512, 1:1,024 and 1:2,048. After incubation at room temperature for 1 h, 400 μl PBS was added to each sample and the mixture was subsequently used to infect VeroE6 cells. After 1.5 h of incubation at room temperature, inoculum was removed and the cells were overlaid with 0.6% Oxoid-agar in DMEM, 20 mM HEPES (pH 7.4), 0.1% NaHCO$_3$, 1% BSA and 0.01% DEAE-Dextran. Cells were fixed 72 h after infection using 4% formaldehyde for 30 min and stained with 1% crystal violet upon removal of the agar overlay. PFU were counted manually. Plaques counted for serum-treated wells were compared to the average number of plaques in the untreated negative controls, which were set to 100%. The PRNT$_{50}$ value was calculated using a linear regression model in GraphPad Prism 9 (GraphPad Prism Software).

### Statistics
Statistical analysis was performed with GraphPad Prism 9 (GraphPad Prism Software). Statistical significance was assessed by one-way ANOVA with a mixed effects model, two-way ANOVA with main effects only, two-tailed Mann–Whitney test with Holm–Šídák multiple comparison, Wilcoxon test and Spearman correlation. Analyses were performed in independent experiments. Statistics was performed for Figs. 1a, b, d, e, 2, Extended Data Figs. 2d, e, 4d, e, f, 5c in $n = 5$ longitudinally analysed vaccines for A*01/S$_{865}$ and A*02/S$_{269}$ and $n = 4$ longitudinally analysed vaccines for A*03/S$_{378}$; for Fig. 3a, b, c, Extended Data Fig. 6e, g in $n = 8$ longitudinally analysed vaccinees for DRB1*15:01/S$_{236}$; for

Fig. 3d, e in 8 longitudinally analysed vaccines; for Fig. 3f, g, Extended Data Fig. 7c in $n = 8$ longitudinally analysed vaccines and $n = 8$ donors with a history of natural SARS-CoV-2 infection cross-sectionally; for Fig. 4a, b, d, Extended Data Fig. 10 in $n = 11$ cross-sectionally analysed vaccinees for $A*01/S_{865}$ at 80–120 dpb, $n = 9$ cross-sectionally analysed vaccinees for $A*02/S_{269}$ at 80-120 dpb, $n = 8$ cross-sectionally analysed vaccinees for $A*03/S_{378}$ at 80–120 dpb, $n = 10$ donors with a history of natural SARS-CoV-2 infection cross-sectionally for $A*01/S_{865}$ 80–120 dpb, $n = 10$ donors with a history of natural SARS-CoV-2 infection cross-sectionally for $A*02/S_{269}$ 80–120 dpb and $n = 10$ donors with a history of natural SARS-CoV-2 infection cross-sectionally for $A*03/S_{378}$ 80-120 dpb; Extended Data Fig. 7b, h in $n = 8$ longitudinally analysed vaccines; Extended Data Fig. 7d in $n = 7$ longitudinally analysed vaccinees; for Extended Data Fig. 7e in $n = 16$ donors with a history of natural SARS-CoV-2 infection cross-sectionally/longitudinally; for Extended Data Fig. 8b, 9b, c in $n = 5$ longitudinally analysed vaccinees for $A*01/S865$ ($n = 5$ at 20–40 dpb and 4 at 40–80 dpb), $n = 5$ longitudinally analysed vaccinees for $A*02/S269$ ($n = 5$ at 20–40 dpb and 4 at 40–80 dpb), $n = 4$ longitudinally analysed vaccinees for $A*03/S378$ ($n = 4$ at 20–40 dpb and 2 at 40–80 dpb), $n = 9$ donors with a history of natural SARS-CoV-2 infection cross-sectionally for $A*01/S865$ ($n = 5$ at 20–40 dpb, 4 at 40–80 dpb), $n = 8$ donors with a history of natural SARS-CoV-2 infection cross-sectionally for $A*02/S269$ ($n = 4$ at 20–40 dpb, 4 at 40–80 dpb), $n = 7$ donors with a history of natural SARS-CoV-2 infection cross-sectionally for $A*03/S378$ ($n = 4$ at 20–40 dpb, 3 at 40–80 dpb). $*P < 0.05; **P < 0.01; ***P < 0.001; ****P < 0.0001$.

## Reporting summary

Further information on research design is available in the Nature Research Reporting Summary linked to this paper.

## Data availability

Patient-related data not included in the paper were generated as part of clinical examination and may be subject to patient confidentiality. Further raw and supporting data conflicting with patient confidentl, are available from the corresponding authors upon request (response within two weeks). Requests for these data will be reviewed by the corresponding authors to verify if the request is subject to any intellectual property or confidentiality obligations. Reference viral sequences SARS-CoV-2 (MN908947.3) https://www.ncbi.nlm.nih.gov/nuccore/MN908947, 229E (NC_002645) https://www.ncbi.nlm.nih.gov/nuccore/NC_002645, HKU1 (NC_006577) https://www.ncbi.nlm.nih.gov/nuccore/NC_006577, NL63 (NC_005831) https://www.ncbi.nlm.nih.gov/nuccore/NC_005831, OC43 (NC_006213) https://www.ncbi.nlm.nih.gov/nuccore/NC_006213, MERS (NC_019843) https://www.ncbi.nlm.nih.gov/nuccore/NC_019843, SARS-CoV-1 (NC_004718) https://www.ncbi.nlm.nih.gov/nuccore/NC_004718) were downloaded from the NCBI database (https://www.ncbi.nlm.nih.gov/). Any data and materials that can be shared will be released via a Material Transfer Agreement. Source data are provided with this paper.

## Code availability

R code to reproduce the analyses of multiparametric flow-cytometry data are available at https://github.com/sagar161286/SARSCoV2_specific_CD8_Tcells.

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

**Acknowledgements** We thank all donors for participating in the current study and the FREEZE-Biobank Center for biobanking (Freiburg University Medical Center) and the Medical Faculty for support. The study was funded by the Federal Ministry of Education and Research (grant number 01KI2077 to G.K., M.H., M.S. and R.T.) and by COVID-19 research grants of the Ministry of Science, Research and Art, State of Baden-Wuerttemberg (COVID-19/AZ.: AZ33-7533-6-10/89/8 to C.N.-H. and B.B.). The presented work was also supported by CRC/TRR 179-Project 01 and CRC 1160-Project A02 (to R.T.), CRC/TRR 179-Project 02 and CRC 1160-Project A06 (to C.N.-H.), CRC 1160-Project B02 (M.R.), CRC/TRR 179-Project 04 (to T.B.), CRC/TRR 179-Project 20 and CRC 1160-Project A02 (to M.H.), CRC/TRR 179-Project 21, CRC 1160-Project A03 and BE-5496/5-1 (to B.B.) of the German Research Foundation (DFG; TRR 179 project no. 272983813; CRC 1160 project no. 256073931). M.H. was supported by a Margarete von Wrangell fellowship (State of Baden-Wuerttemberg). T.B. was supported by the Berta-Ottenstein Programme, Faculty of Medicine, University of Freiburg. H.L. was supported by the IMM-PACT-Programme for Clinician Scientists, Department of Medicine II, Medical Center – University of Freiburg and Faculty of Medicine, University of Freiburg. The funding body had no role in the decision to write or submit the manuscript.

**Author contributions** V.O., H.L., J.K. and I.S. planned, performed and analysed experiments with the help of K.C., S.G., B.C., I.J., J.S., K.W., S.M., J.L.M., J.M.L., K.B., J.F., F.T., A.J., K.Z. S. and O.S. H.L., J.L.M., B.B., C.W., H.H., K.S. and S.R. were responsible for donor recruitment. F.E. performed four-digit HLA typing by next-generation sequencing. T.B., K.Z., G.K., M.S., M.R., R.T., C.N.-H. and M.H. contributed to experimental design, planning and supervision. V.O., H.L., J.K., I.S., R.T., M.H. and C.N.-H. interpreted data and wrote the manuscript. M.H., C.N.-H. and R.T. designed the study and are joint last authors.

**Competing interests** The authors declare no competing interests.

**Additional information**
**Correspondence and requests for materials** should be addressed to R.T., C.N.-H. or M.H.

**a**

prime boost vaccination

analysis

BL 0 3-4 6-8 9-12 13-15 16-18 19-21 0 2-4 5-6 7-9 13-16 28-30 55-58 80-120

blood/serum collection

dpp dpb

**b**

Lymphocytes — 60% ; 98%

Single cells 2 ; Single cells 1 — 30% ; 97%

CD8+ — 22%

**c**

A*01/S₈₆₅ — L T D E M I A Q Y

| | |
|---|---|

A*02/S₂₆₉ — Y L Q P R T F L

A*03/S₃₇₈ — K C Y G V S P T K

Upper panel (SARS-CoV-1/2, MERS, HKU1, NL63, OC43, 229E):

A*01/S865 positions 865–873
- SARS-CoV-2: . . . . . . . . .
- SARS-CoV-1 (847–855): · · D · · · A ·
- MERS (939–947): M D V N · E · A ·
- HKU1 (954–962): · S E S Q · S G ·
- NL63 (921–929): A D A · R M · M ·
- OC43 (953–961): · S E N Q · S G ·
- 229E (740–748): A D A · R M · M ·

A*02/S269 positions 269–277
- SARS-CoV-2: . . . . . . . .
- SARS-CoV-1 (256–): · · K · T · M
- MERS (317–): K · · L · · ·
- HKU1 (260–): P · S K · Q Y
- NL63 (265–): G · K S S · G F
- OC43 (268–276): P · T S · O Y
- 229E (87–95): G · R F T · G F V

A*03/S378 positions 378–386
- SARS-CoV-2: . . . . . . . . .
- SARS-CoV-1 (365–373): · · · A · · · ·
- MERS (369–377): T · S Q I · · A A
- HKU1 (425–431): S · N N F D E S ·
- NL63 (377–385): V · – N · · A · N
- OC43 (242–248): T · N N I D A A ·
- 229E (–): V · – N · · Q · S

Lower panel (VOC):

A*01/S₈₆₅ — L T D E M I A Q Y (865–873)
A*02/S₂₆₉ — Y L Q P R T F L L (269–277)
A*03/S₃₇₈ — K C Y G V S P T K (378–386)

- B.1 (865–873 / 269–277 / 378–386): . . . . . . . . .
- Alpha (862–870 / 266–274 / 375–383): . . . . . . . . .
- Beta (862–870 / 266–274 / 375–383): . . . . . . . . .
- Gamma (865–873 / 269–277 / 378–386): . . . . . . . . .
- Delta (865–873 / 269–277 / 378–386): . . . . . . . . .

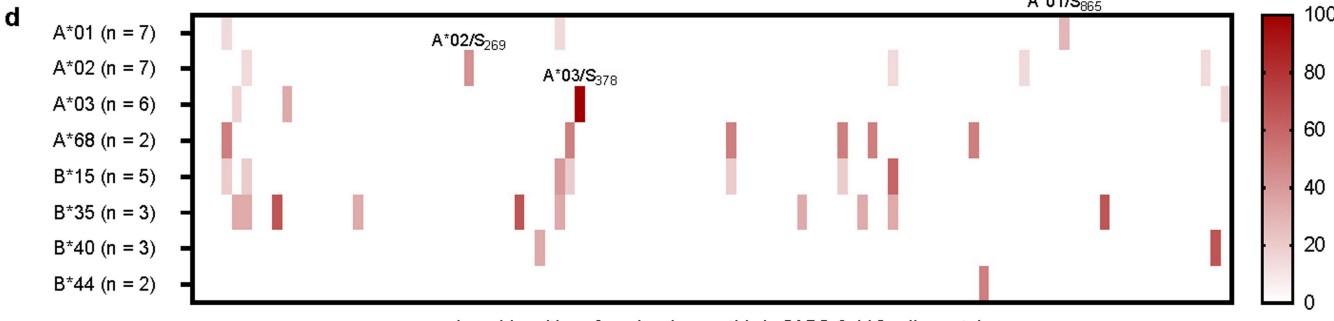

| Epitope | Predicted IC₅₀ (ANN 4.0) | NetCTLpan (%Rank) | NetMHCstabpan | | |
|---|---|---|---|---|---|
| | | | Thalf(h) | %Rank_Stab | BindLevel |
| A*01/S₈₆₅ | 3.37 | 0.01 | 11.71 | 0.01 | strong binder |
| A*02/S₂₆₉ | 5.36 | 0.05 | 8.04 | 0.3 | strong binder |
| A*03/S₃₇₈ | 152.62 | 1.5 | 1.21 | 0.8 | weak binder |

**d**

A*01 (n = 7)
A*02 (n = 7)
A*03 (n = 6)
A*68 (n = 2)
B*15 (n = 5)
B*35 (n = 3)
B*40 (n = 3)
B*44 (n = 2)

A*02/S₂₆₉ A*03/S₃₇₈ A*01/S₈₆₅

aminoacid position of overlapping peptide in SARS-CoV-2 spike protein

% — 0 20 40 60 80 100

**Extended Data Fig. 1 | Spike-specific CD8+ T cell epitopes following vaccination. (a)** Timeline showing blood and serum collection before and after prime and boost vaccination. **(b)** Gating strategy of flow cytometry data. A*01/S₈₆₅-, A*02/S₂₆₉- and A*03/S₃₇₈-specific CD8+ T cells were identified via pMHCI tetramer-based analysis. **(c)** Comparison of epitope sequences with amino acid sequences of SARS-CoV-1/2, MERS and common cold coronaviruses amino acid sequences (upper panel) and with circulating SARS-CoV-2 variants of concern (VOC) (middle panel), respectively for A*01/S₈₆₅, A*02/S₂₆₉ and A*03/S₃₇₈-specific CD8+ T cell epitopes. (Lower panel) A*01/S₈₆₅, A*02/S₂₆₉ and A*03/S₃₇₈ peptide characteristics, comparing different prediction methods for the estimation of MHC I binding affinity, half-life and processing. **(d)** Heatmap showing the percentage of patients with a CD8+ T cell response to spike overlapping peptides/fine-mapped minimal optimal epitope in relation to the total number of patients tested with the respective HLA type. *n* = 16 OLP: overlapping peptide. Parts of the figure were drawn by using pictures from Servier Medical Art (http://smart.servier.com/) and licensed under a Creative Common Attribution 3.0 Generic License (a).

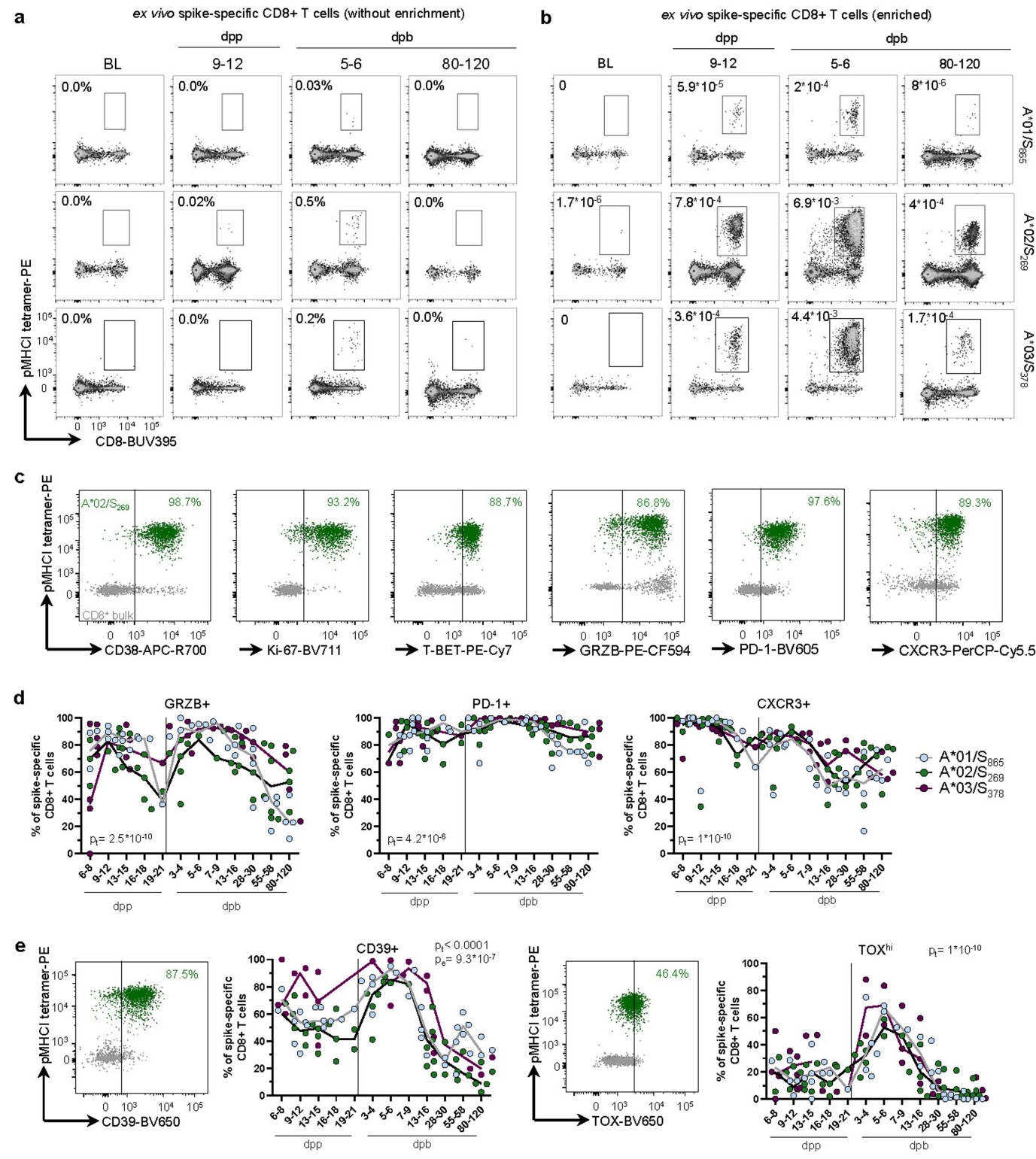

**Extended Data Fig. 2 | Spike-specific CD8+ T cells before and after enrichment. (a, b)** Dot plots showing A*01/$S_{865}$-, A*02/$S_{269}$- and A*03/$S_{378}$-specific CD8+ T cells ex vivo without pMHCI tetramer-based enrichment **(a)** and after pMHCI tetramer-based enrichment **(b)** at BL, before and after boost vaccination. **(c)** Exemplary dot plots (5-6 dpb) depicting the expression levels of CD38, Ki-67, T-BET, GRZB, PD-1 and CXCR3 in A*02/$S_{269}$- (green) specific and bulk (grey) CD8+ T cells. **(d)** % of GRZB, PD-1 and CXCR3 expressing A*01/$S_{865}$-, A*02/$S_{269}$- and A*03/$S_{378}$-specific non-naïve CD8+ T cells.

**(e)** Exemplary dot plot (5-6 dpb) depicting the expression levels of CD39 and TOX in A*02/$S_{269}$- (green) specific and bulk (grey) CD8+ T cells. % expression among A*01/$S_{865}$-, A*02/$S_{269}$-and A*03/$S_{378}$-specific non-naïve CD8+ T cells is shown on the right side. BL: baseline; dpp: days post prime; dpb: days post boost; GRZB: granzyme B. (d-e) Two-way ANOVA with main effects only comparing the effect of the different epitopes and of time course. All statistically significant results are marked with the respective exact p-value ($p_e$: epitope, $p_t$: time).

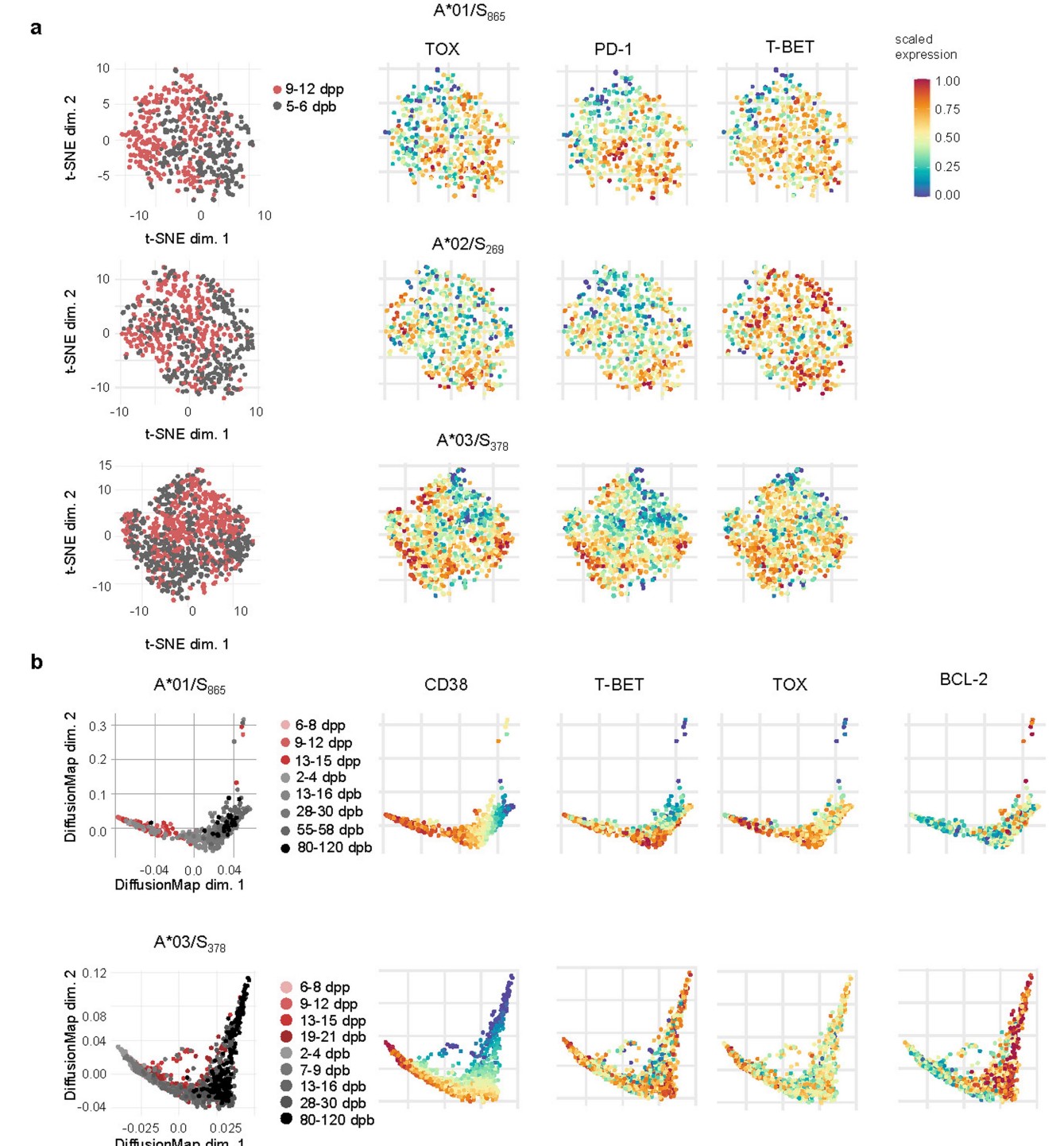

**Extended Data Fig. 3 | Expression levels of CD38, T-BET, TOX, PD-1 and BCL-2 after prime and boost vaccination. (a)** t-SNE representation of flow cytometry data comparing A*01/$S_{865}$-, A*02/$S_{269}$- and A*03/$S_{378}$-specific CD8+ T cells after prime and boost vaccination (prime 4, 3 and 5 and boost 3, 3 and 5 individuals for A*01/$S_{865}$, A*02/$S_{269}$ and A*03/$S_{378}$, respectively). Expression levels of TOX, PD-1 and T-BET are indicated for A*01/$S_{865}$, A*02/$S_{269}$ and A*03/$S_{378}$ (colour-code: blue, low expression; red, high expression). **(b)** Diffusion map showing flow cytometry data for A*01/$S_{865}$- and A*03/$S_{378}$-specific CD8+ T cells of one representative donor at dpp (shades of red) and dpb (shades of grey) with CD38, T-BET, TOX and BCL-2 expression levels plotted on the diffusion map (colour-code: blue, low expression; red, high expression). dpp: days post prime; dpb: days post boost; t-SNE: t-distributed stochastic neighbour embedding.

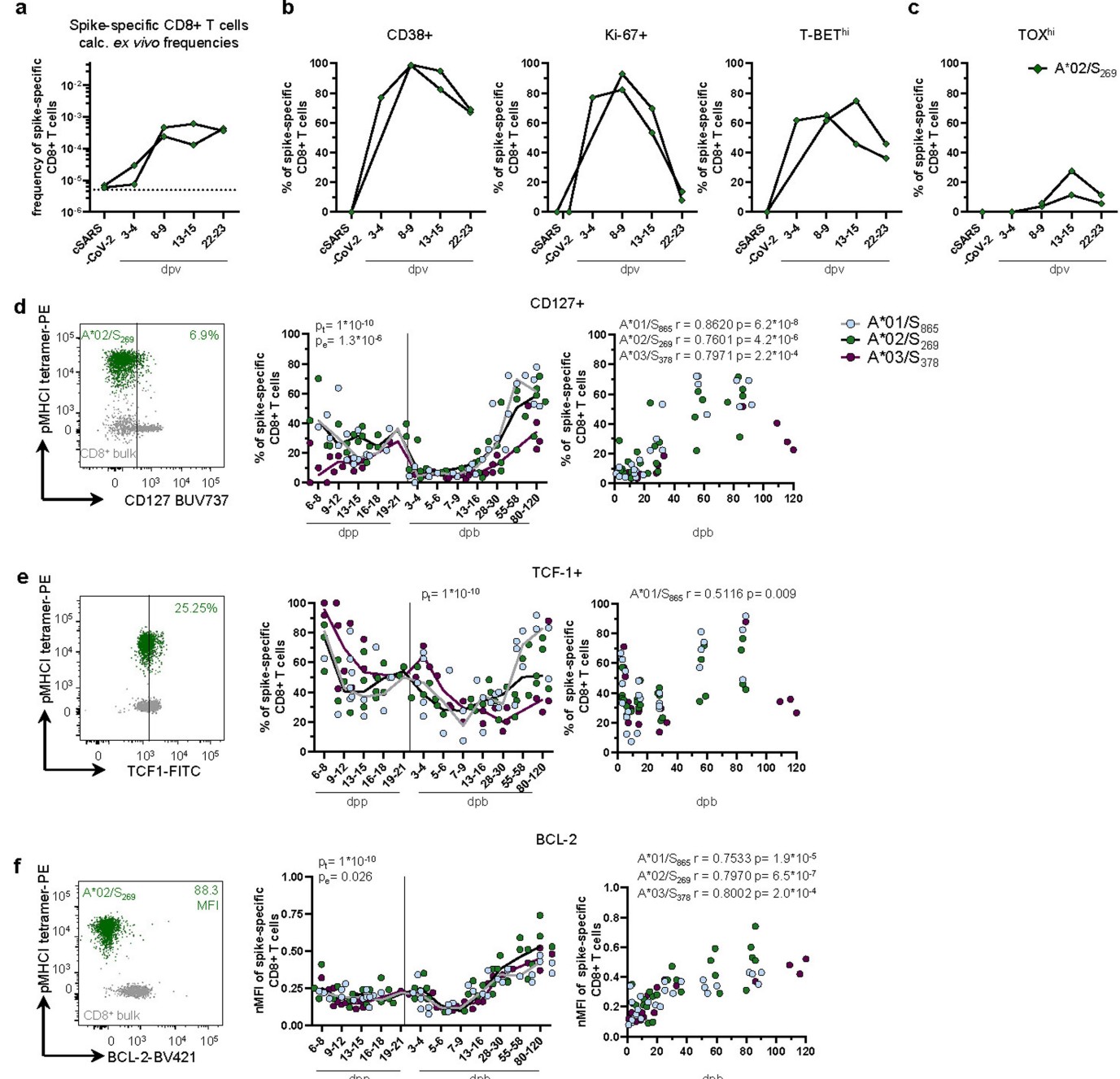

**Extended Data Fig. 4 | Spike-specific CD8+ T cells in SARS-CoV-2 convalescents after a single vaccination and expression of early memory marker after prime and boost vaccination. (a-c)** Data of two donors who recovered from SARS-CoV-2 infection >365 day ago and received a single dose Bnt162b2 vaccination. **(a)** The calculated ex vivo frequency for A*02/$S_{269}$-specific CD8+ T cells in two donors is indicated at baseline (cSARS-CoV-2) and days after single dose vaccination. % of CD38, Ki-67, T-BET[hi] **(b)** and TOX[hi] **(c)** expressing A*02/$S_{269}$- specific non-naïve CD8+ T cells. **(d-e)** Exemplary dot plots depicting the expression levels of CD127 **(d)** and TCF-1 **(e)** in A*02/$S_{269}$- (green) specific and bulk (grey) CD8+ T cells 6 dpb. % A*01/$S_{865}$-, A*02/$S_{269}$- and A*03/$S_{378}$-specific non-naïve CD8+ T cells expressing CD127 **(d)** and TCF-1 **(e)** were determined. Correlation of indicated marker with dpb is depicted on the

right. **(f)** Exemplary dot plots depicting the expression levels of BCL-2 in A*02/$S_{269}$- (green) specific as well as on bulk (grey) CD8+ T cells 6 dpb. nMFI (MFI normalized to naïve CD8+ T cells) of A*01/$S_{865}$-, A*02/$S_{269}$- and A*03/$S_{378}$-specific non-naïve CD8+ T cells expressing BCL-2 was determined. Correlation of nMFI BCL-2 with dpb is depicted on the right. cSARS: convalescent SARS; dpv: days post vaccination; BL: baseline; dpp: days post prime; dpb: days post boost; nMFI: normalized Median Fluorescent Intensity, epi: epitope. (d-f) Two-way ANOVA with main effects only comparing the effect of the different epitopes and of time course (left) and Spearman correlation (right) was performed. All statistically significant results are marked with the respective exact p-value ($p_e$: epitope, $p_t$: time).

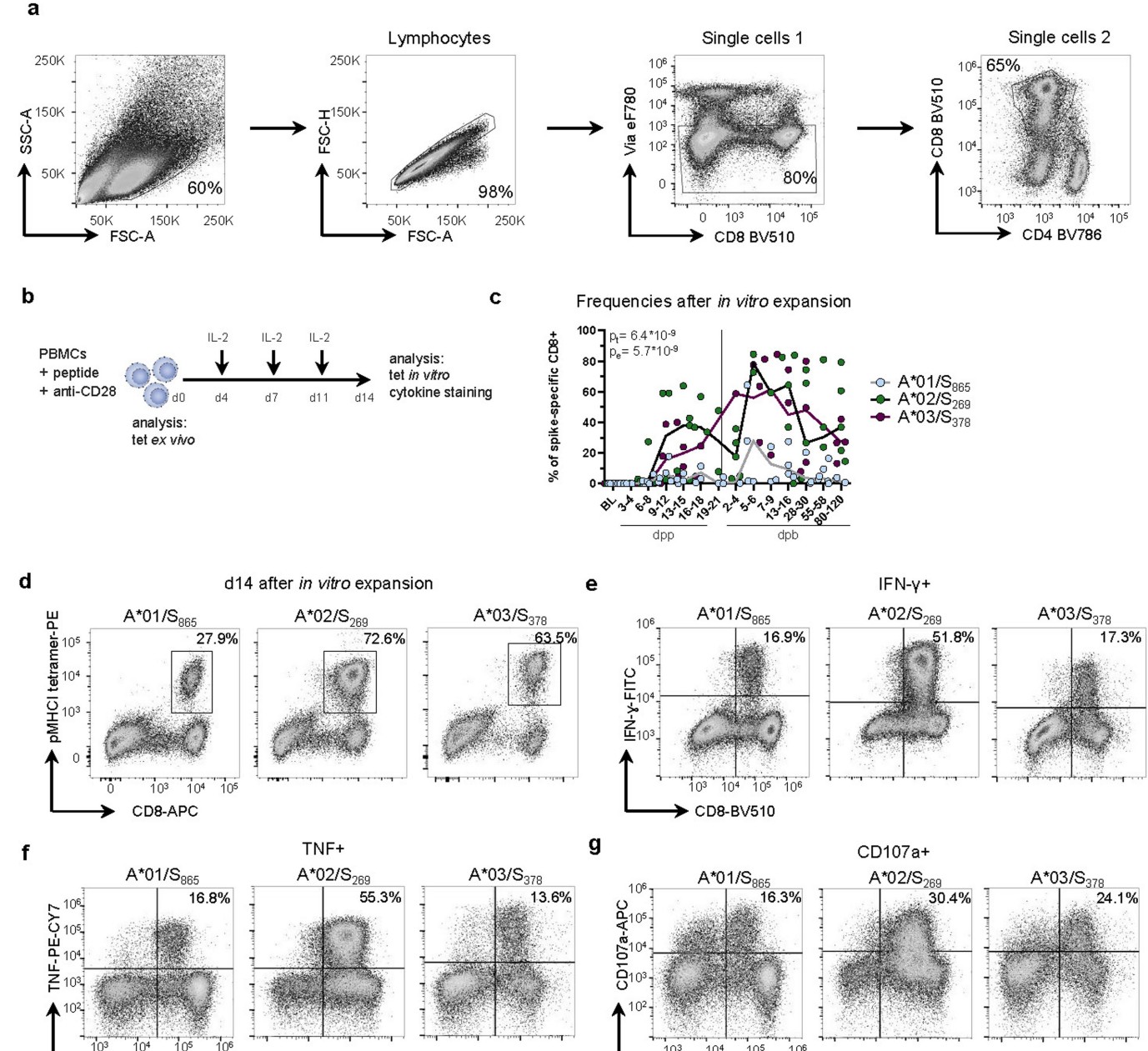

**Extended Data Fig. 5 | Frequency and functional capacity of spike-specific CD8+ T cells following prime and boost vaccination. (a)** Gating strategy of flow cytometry data. Cytokine secretion of A\*01/$S_{865}$-, A\*02/$S_{269}$- and A\*03/$S_{378}$-specific CD8+ T cells was determined after in vitro expansion. **(b)** Workflow depicting peptide-specific in vitro expansion of CD8+ T cells. **(c)** Frequency of A\*01/$S_{865}$-, A\*02/$S_{269}$- and A\*03/$S_{378}$-specific CD8+ T cells after 14 day of in vitro expansion. **(d-g)** Dot plots showing A\*01/$S_{865}$-, A\*02/$S_{269}$- and A\*03/$S_{378}$-specific

CD8+ T cells after in vitro expansion **(d)** and IFN-γ- **(e)**, TNF- **(f)** and CD107a- **(g)** producing CD8+ T cells after in vitro expansion (5-6 dpb). (c) Two-way ANOVA with main effects only comparing the effect of the different epitopes and of time course. BL: baseline; dpp: days post prime; dpb: days post boost, epi: epitope. Two-way ANOVA with main effects only comparing the effect of the different epitopes and of time course. All statistically significant results are marked with the respective exact p-value ($p_e$: epitope, $p_t$: time).

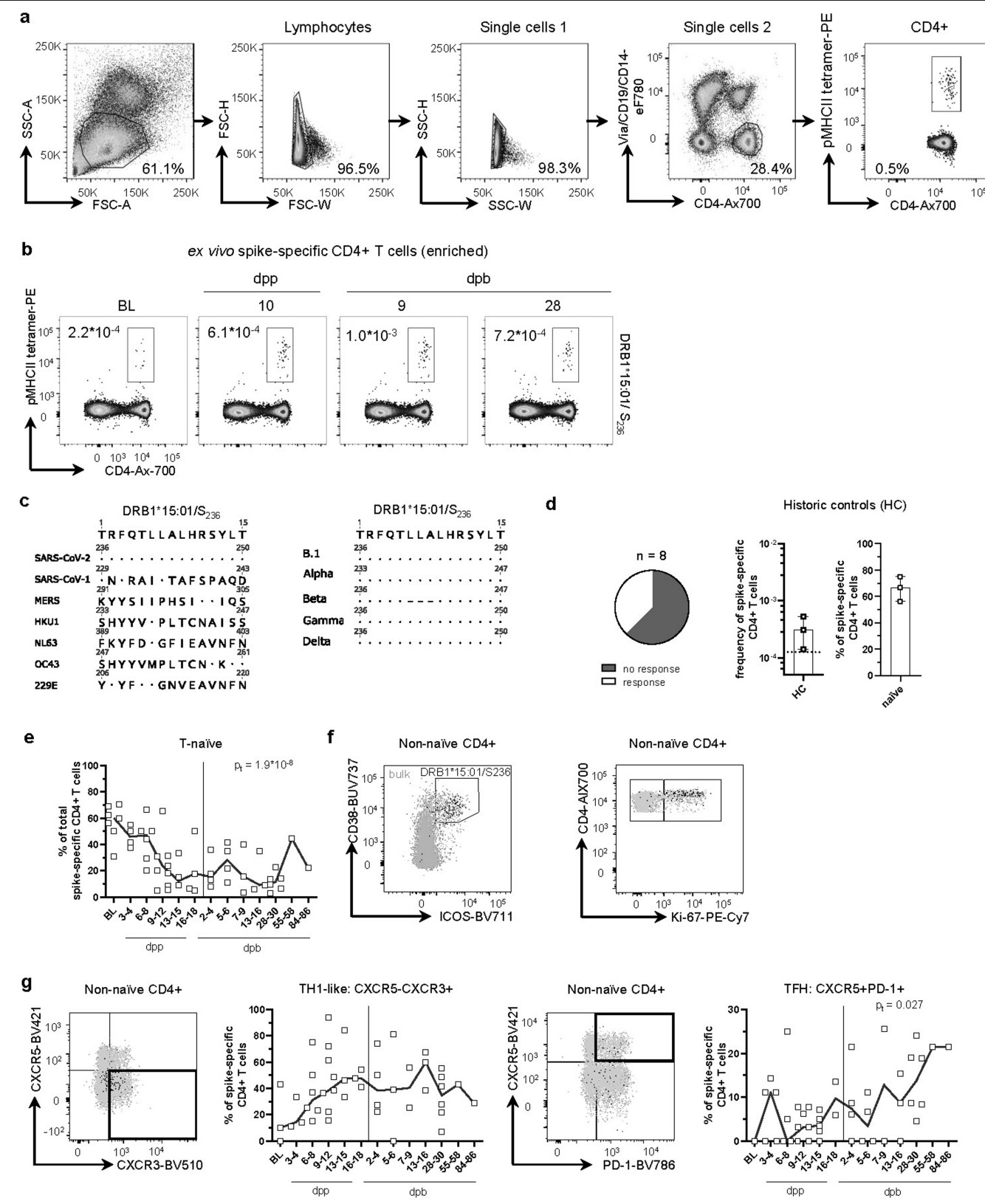

**Extended Data Fig. 6 | See next page for caption.**

**Extended Data Fig. 6 | Circulating spike-specific CD4+ T cells following prime and boost vaccination. (a)** Gating strategy of flow cytometry data. DRB1*15:01/$S_{236}$-specific CD4+ T cells were identified via pMHCII tetramer-based enrichment used in further analyses. **(b)** Dot plots showing DRB1*15:01/$S_{236}$-specific CD4+ T cells ex vivo after pMHCII tetramer-based enrichment at BL, before and after boost vaccination. **(c)** Comparison of DRB1*15:01/$S_{236}$ epitope sequence with amino acid sequences of SARS-CoV-1/2, MERS and common cold coronaviruses (left) and of circulating SARS-CoV-2 variants of concern (VOC) (right). **(d)** Number of responses (left) and the calculated ex vivo frequencies (middle) of DRB1*15:01/$S_{236}$-specific CD4+ T cells in historic controls (HC, $n = 8$). % naïve within total DRB1*15:01/$S_{236}$-specific CD4+ T cells in historic controls (right). Detection limit: $1.25*10^{-4}$. **(e)** % naïve of total DRB1*15:01/$S_{236}$-specific CD4+ T cells. **(f)** Representative dot plots of ICOS+CD38++ and Ki-67+ expression (grey: bulk, black: DRB1*15:01/$S_{236}$-specific CD4+ T cells). **(g)** % CXCR5-CXCR3+ TH1 cells and of CXCR5+PD-1+ TFH within non-naïve DRB1*15:01/$S_{236}$-specific CD4+ T cells with representative dot plots (grey: bulk, black: DRB1*15:01/$S_{236}$-specific CD4+ T cells). BL: baseline, dpp: days post prime, dpb: days post boost; TFH: follicular helper T cells; TH1: T helper cells 1-like cells. Bar charts show the median with IQR. (e, g) One-way ANOVA with a mixed effects model comparing the effect of the time course. All statistically significant results are marked with the respective exact p-value ($p_t$: time).

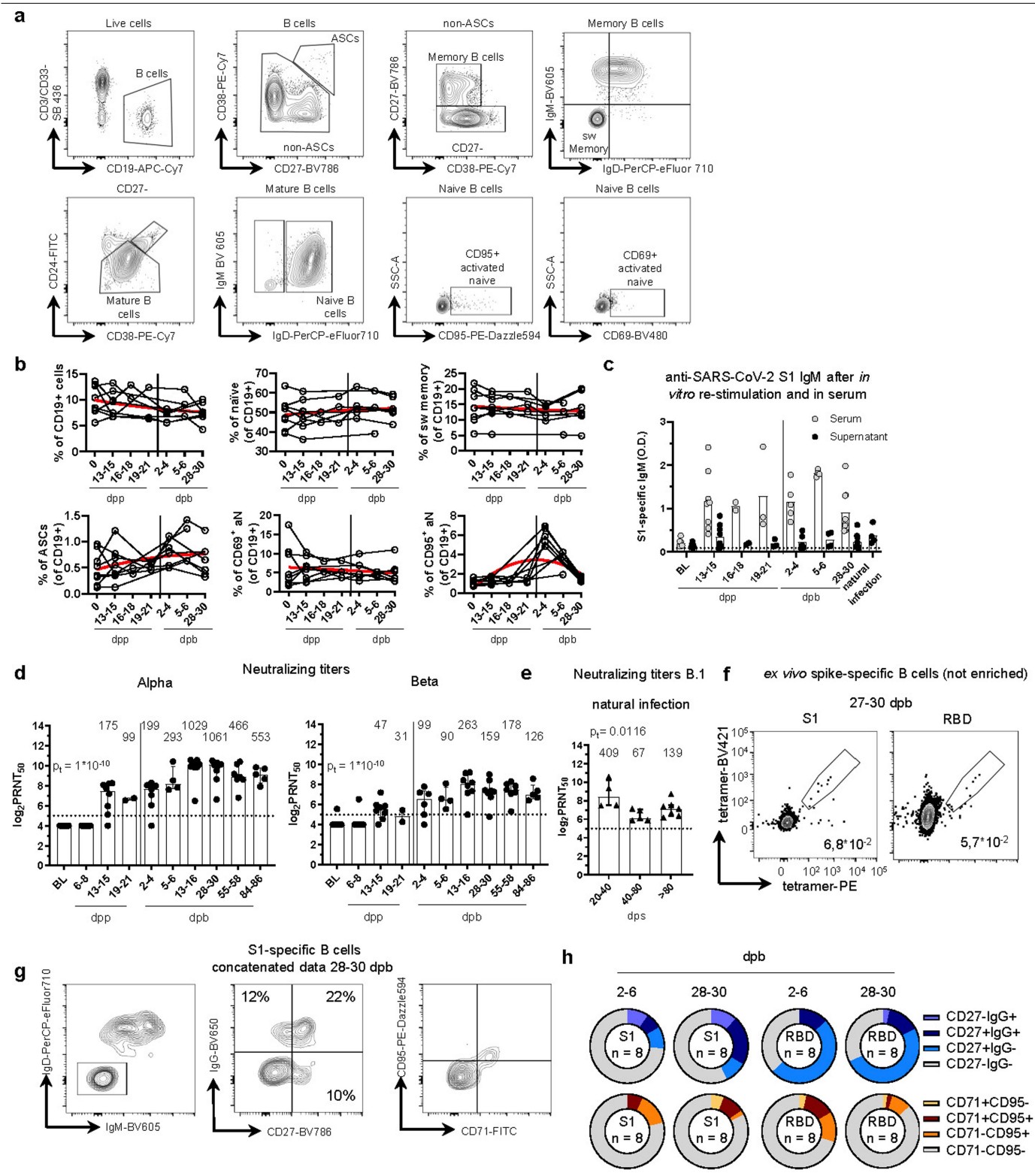

**Extended Data Fig. 7** | See next page for caption.

**Extended Data Fig. 7 | Circulating B cells, antibodies and antibody neutralization activity after prime and boost vaccination. (a)** Gating strategy of flow cytometry data for different B cell subpopulations. **(b)** % of CD19+, naïve, switched memory, ASCs, CD69+aN and CD95+ aN cells within bulk B cells was determined at BL, post prime/boost. Nonlinear fit was calculated in red. **(c)** Detection of spike IgM in serum at BL, post prime/boost by ELISA. Level of secreted IgM was determined in supernatant of PBMC from BL, post prime/boost and after natural infection after in vitro stimulation for 9 d with CpG and IL-2. Detection limit: O.D. $8.5*10^{-2}$. **(d)** Antibody neutralization activity is depicted as $PRNT_{50}$ at BL, dpp and dpb vaccination for the SARS-CoV-2 VOC alpha and beta. **(e)** Antibody neutralization capacity is depicted as $PRNT_{50}$ at different dps in natural SARS-CoV-2 infection with SARS-CoV-2 ancestral variant B.1. (d-e) Numbers indicate non-logarithmic median value. Detection limit: $5 \log_2 PRNT_{50}$. **(f)** Dot plots showing double tetramer positive B cells for S1 and RBD epitope at 28-30 dpb. **(g)** Dot plots representing co-expression of IgD/IgM (left), IgG/CD27 (middle) and CD95/CD71 (right) on concatenated flies of S1-specific B cells at 28-30 dpb. **(h)** Donut plots representing co-expression of CD27/IgG (upper) and CD71/CD95 (lower) in concatenated analysis of S1-specific B cells populations (n indicated the number of individual files concatenated) after boost vaccination. sw memory: switched memory; aN: activated naïve; BL: baseline; dpp: days post prime; dpb: days post boost; dps: days post symptoms; O.D.: optical density; $PRNT_{50}$: plaque-reduction neutralization titer 50. Bar charts show the median with IQR. (c-e) One-way ANOVA with a mixed effects model comparing the effect the time course. All statistically significant results are marked with the respective exact p-value ($p_t$: time).

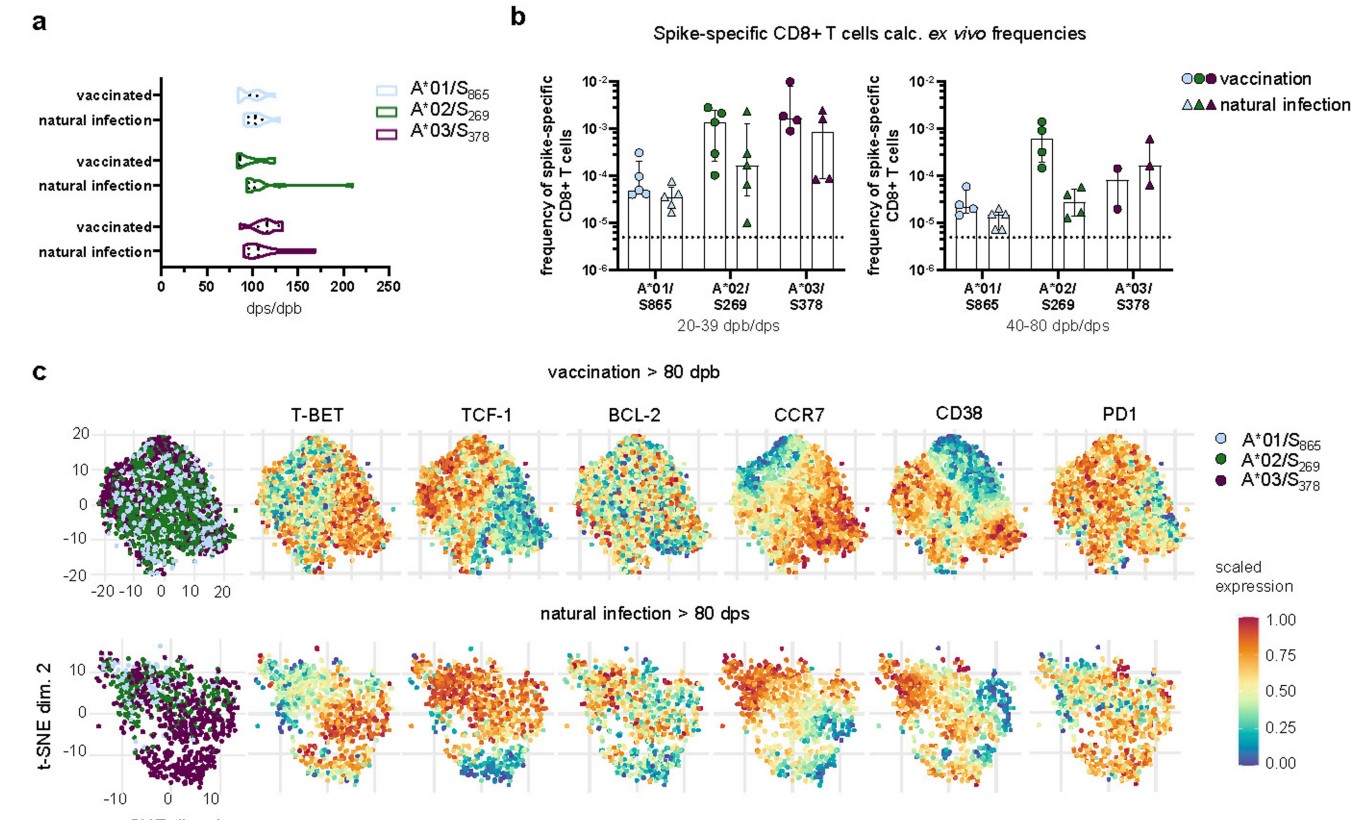

**Extended Data Fig. 8 | Comparison of spike-specific CD8+ T cells after vaccination and natural infection. (a)** Distribution of dpb and dps of donors analysed for A*01/$S_{865}$-, A*02/$S_{269}$- and A*03/$S_{378}$-specific CD8+ T cells after vaccination and natural infection. **(b)** Calculated frequency of A*01/$S_{865}$-, A*02/$S_{269}$- and A*03/$S_{378}$-specific CD8+ T cells ex vivo after pMHCI tetramer-based enrichment at 20-39 and 40-80 dpb/dps. Detection limit: $5*10^{-6}$. **(c)** Plotted expression levels of T-BET, TCF-1, BCL-2, CCR7, CD38 and PD1 on t-SNE depicting all spike-specific CD8+ T cells >80 dpb vaccination (upper) and >80 dps in natural SARS-CoV-2 infection (lower) are indicated for A*01/$S_{865}$, A*02/$S_{269}$ and A*03/$S_{378}$ (colour-code: blue, low expression; red, high expression). dpb: days post boost; dps: days post symptom onset; t-SNE: t-distributed stochastic neighbour embedding. Bar charts show the median with IQR. (a) 2-way ANOVA including Tukey's multiple comparisons test were performed. (b) Statistical analyses of vaccination vs. natural infection was performed by Mann–Whitney test with Holm-Šídák method. All statistically significant results are marked with the respective exact p-value.

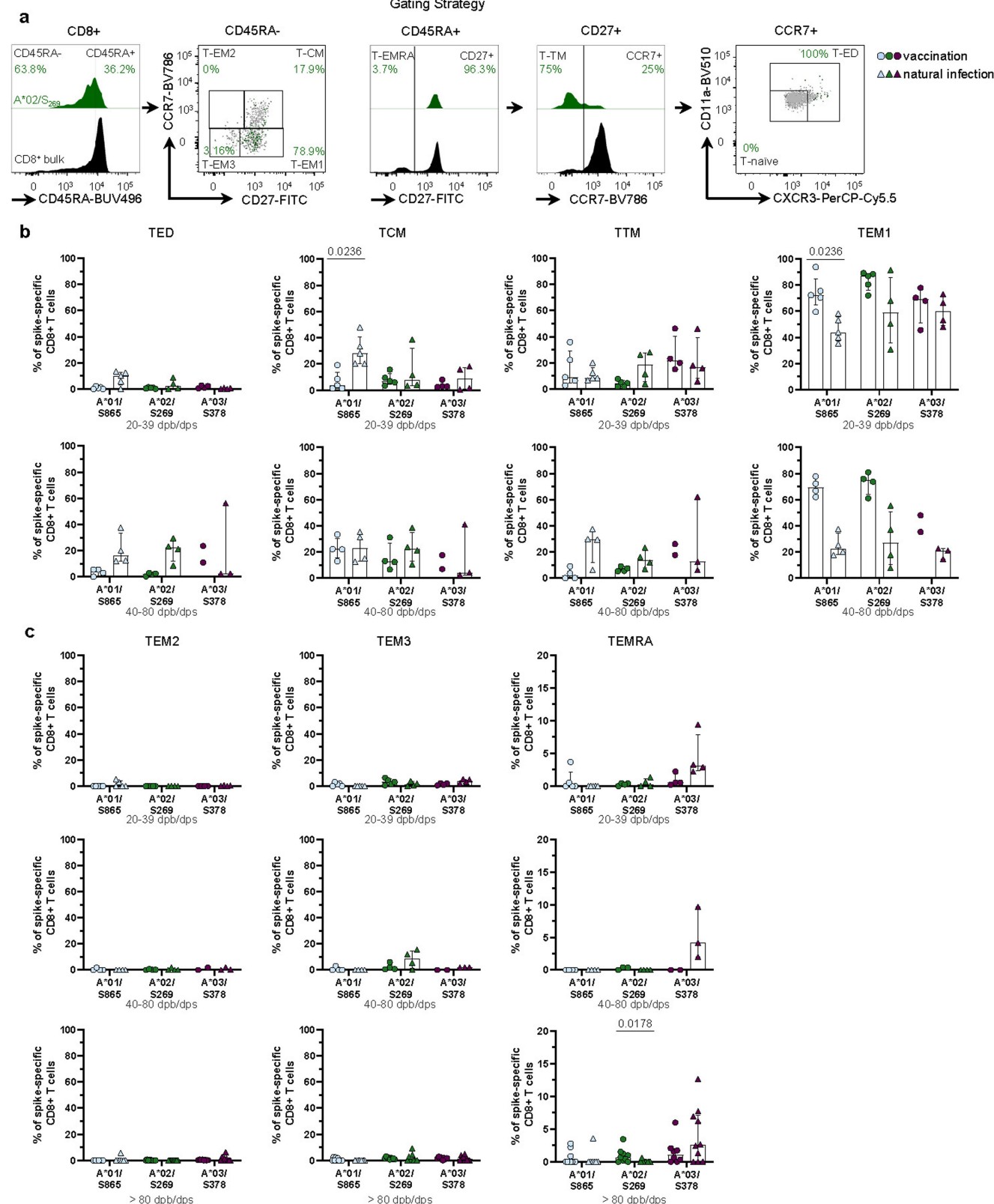

**Extended Data Fig. 9 | Subset distribution of spike-specific CD8+ T cell after vaccination and natural infection. (a)** Gating strategy of memory CD8+ T cell populations among A*02/S₂₆₉⁻ (green) (36 dps) specific CD8+ T cells, (grey/black: bulk CD8+ T cells). **(b)** Distribution of spike-specific CD8+ T-cell memory subsets TED, TCM, TTM and TEM1 at 20-39 and 40-80 dpb/dps. **(c)** Distribution of spike-specific CD8+ T-cell memory subsets TEM2, TEM3 and TEMRA at 20-39, 40-80 and >80 dpb/dps. dpb: days post boost; dps: days post symptoms; ; TED: early differentiated, TCM: central memory T cells and TEM1: effector memory T cells 1; TEM2: effector memory T cells 2; TEM3: effector memory T cells 3, TEMRA terminally differentiated effector memory cells re-expressing CD45RA. Bar charts show the median with IQR. (b-c) Statistical analyses of vaccination vs. natural infection was performed by Mann–Whitney test with Holm-Šídák method. All statistically significant results are marked with the respective exact p-value.

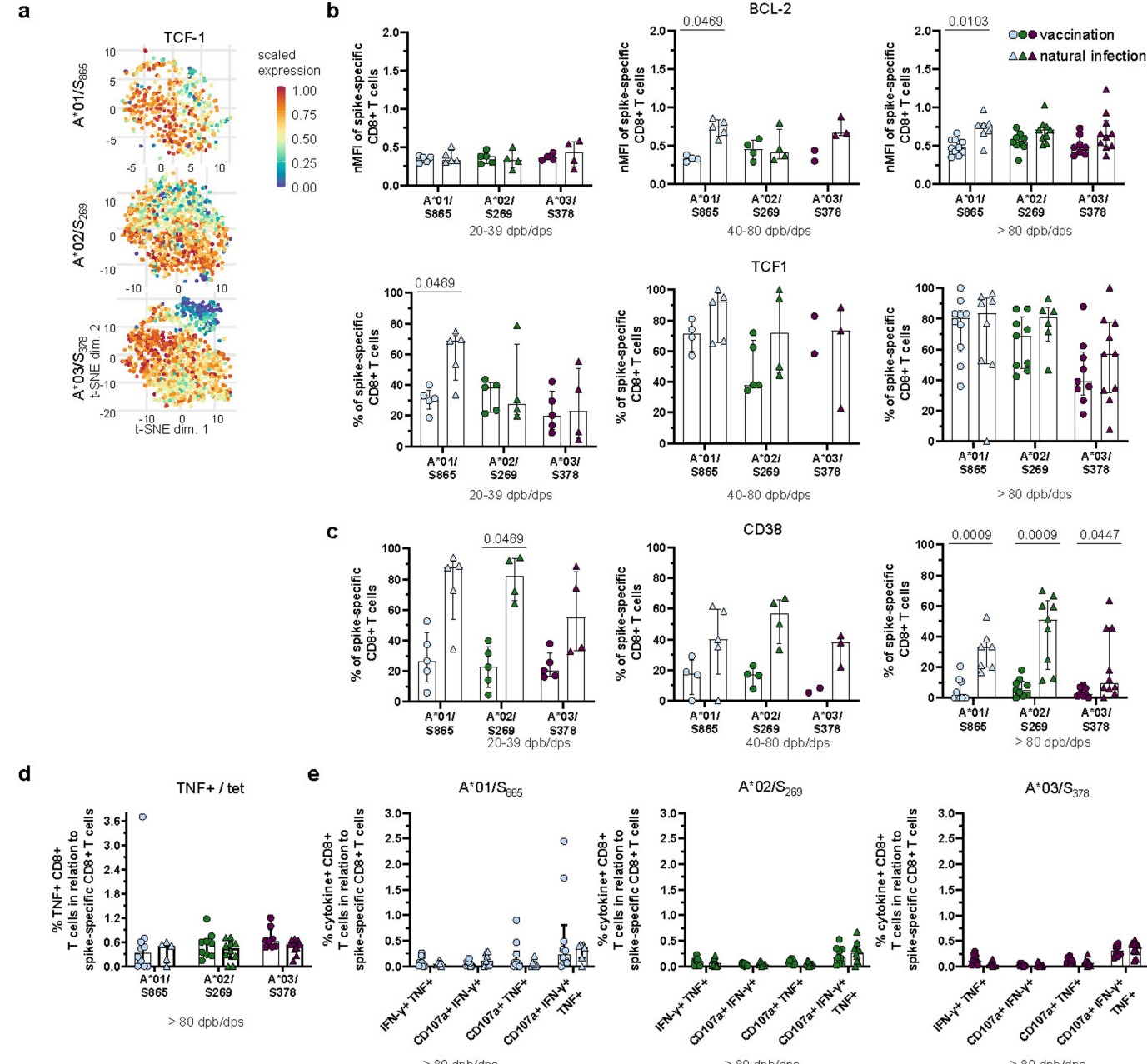

**Extended Data Fig. 10 | Expression levels of BCL-2, TCF1 and CD38 and polyfunctionality of spike-specific CD8+ T cells after vaccination and natural infection. (a)** Plotted expression levels of TCF-1 on t-SNE depicting spike-specific CD8+ T cells >80 dpb vaccination and dps of natural infection for A*01/S$_{865}$- (vaccination $n = 9$, natural infection $n = 9$), A*02/S$_{269}$-(vaccination $n = 10$, natural infection $n = 8$) and A*03/S$_{378}$- (vaccination $n = 9$, natural infection $n = 9$) (colour-code: blue, low expression; red, high expression). **(b, c)** nMFI (normalized to naïve CD8+ T cells) of BCL-2 (b, upper), TCF1 (b, middle) and CD38 (c, lower) spike-specific non-naïve CD8+ T cells at 20-39, 40-80 and

>80 dpb/dps. **(d)** % of TNF-producing CD8+ T cells in relation to the frequency of spike-specific CD8+ T cells after in vitro expansion. **(e)** Bar graphs depicting the polyfunctionality of the respective spike-specific CD8+ T cells comparing vaccination and natural infection. dpb: days post boost; dps: days post symptom onset. Bar charts show the median with IQR. (b-e) Statistical analyses of vaccination vs. natural infection was performed by Mann–Whitney test with Holm–Šídák method. All statistically significant results are marked with the respective exact p-value.

Maike Hofmann
Christoph Neumann-Haefelin

# Reporting Summary

Nature Research wishes to improve the reproducibility of the work that we publish. This form provides structure for consistency and transparency in reporting. For further information on Nature Research policies, see our Editorial Policies and the Editorial Policy Checklist.

## Statistics

For all statistical analyses, confirm that the following items are present in the figure legend, table legend, main text, or Methods section.

| n/a | Confirmed | |
|---|---|---|
| ☐ | ☒ | The exact sample size (*n*) for each experimental group/condition, given as a discrete number and unit of measurement |
| ☐ | ☒ | A statement on whether measurements were taken from distinct samples or whether the same sample was measured repeatedly |
| ☐ | ☒ | The statistical test(s) used AND whether they are one- or two-sided *Only common tests should be described solely by name; describe more complex techniques in the Methods section.* |
| ☐ | ☒ | A description of all covariates tested |
| ☐ | ☒ | A description of any assumptions or corrections, such as tests of normality and adjustment for multiple comparisons |
| ☐ | ☒ | A full description of the statistical parameters including central tendency (e.g. means) or other basic estimates (e.g. regression coefficient) AND variation (e.g. standard deviation) or associated estimates of uncertainty (e.g. confidence intervals) |
| ☐ | ☒ | For null hypothesis testing, the test statistic (e.g. *F*, *t*, *r*) with confidence intervals, effect sizes, degrees of freedom and *P* value noted *Give P values as exact values whenever suitable.* |
| ☐ | ☒ | For Bayesian analysis, information on the choice of priors and Markov chain Monte Carlo settings |
| ☐ | ☒ | For hierarchical and complex designs, identification of the appropriate level for tests and full reporting of outcomes |
| ☐ | ☒ | Estimates of effect sizes (e.g. Cohen's *d*, Pearson's *r*), indicating how they were calculated |

*Our web collection on statistics for biologists contains articles on many of the points above.*

## Software and code

Policy information about availability of computer code

Data collection   All software used to perform data collection are described in the methods section of the manuscript or the supportive information. Multiparametric Flow cytometry data was collected on FACSCanto II, LSRFortessa with FACSDiva software version 10.6.2 (BD, Germany) or CytoFLEX (Beckman Coulter) with CytExpert Software version 2.3.0.84 or Cytek Aurora (Cytek Biosciences) with SpectroFlo® Software version 2.2.0.3. ELISA data was collected by SparkControl magellan software version 2.2.

Data analysis   All codes used to perform bioinformatic analyses are described in the methods section of the manuscript or the supportive information. Multiparametric flow cytometry data was analyzed using FlowJo software version 10.6.2 (Treestar, Becton Dickinson). The visualization of multiparametric flow cytometry data was done with R version 4.0.2 using the Bioconductor (version: Release (3.11)) CATALYST package (Crowell H, Zanotelli V, Chevrier S, Robinson M (2020). CATALYST: Cytometry dATa anALYSis Tools. R package version 1.12.2, https://github.com/HelenaLC/CATALYST). R code to reproduce the analyses of multiparametric flow-cytometry data is available at https://github.com/sagar161286/SARSCoV2_specific_CD8_Tcells. Visualization and statistical analysis was performed using GraphPad 9 software. Sequence homology analyses were performed in Geneious Prime 2020.0.3 (https://www.geneious.com/) using Clustal Omega 1.2.2 alignment with default settings. Reference viral sequences SARS-CoV-2 (MN908947.3) https://www.ncbi.nlm.nih.gov/nuccore/MN908947, 229E (NC_002645) https://www.ncbi.nlm.nih.gov/nuccore/NC_002645, HKU1 (NC_006577) https://www.ncbi.nlm.nih.gov/nuccore/NC_006577, NL63 (NC_005831) https://www.ncbi.nlm.nih.gov/nuccore/NC_005831, OC43 (NC_006213) https://www.ncbi.nlm.nih.gov/nuccore/NC_006213, MERS (NC_019843) https://www.ncbi.nlm.nih.gov/nuccore/NC_019843, SARS-CoV-1 (NC_004718) https://www.ncbi.nlm.nih.gov/nuccore/NC_004718) were downloaded from the NCBI database (https://www.ncbi.nlm.nih.gov/).

For manuscripts utilizing custom algorithms or software that are central to the research but not yet described in published literature, software must be made available to editors and reviewers. We strongly encourage code deposition in a community repository (e.g. GitHub). See the Nature Research guidelines for submitting code & software for further information.

## Data

Policy information about availability of data

All manuscripts must include a data availability statement. This statement should provide the following information, where applicable:
- Accession codes, unique identifiers, or web links for publicly available datasets
- A list of figures that have associated raw data
- A description of any restrictions on data availability

Raw data in this study are provided in the Source data. Additional supporting data are available from the corresponding authors upon reasonable request. All requests for raw and analyzed data and materials will be reviewed by the corresponding authors to verify if the request is subject to any intellectual property or confidentiality obligations. Patient-related data not included in the paper were generated as part of clinical examination and may be subject to patient confidentiality. Any data that can be shared will be released via a Material Transfer Agreement.
Reference viral sequences 229E (NC_002645), HKU1 (NC_006577), NL63 (NC_005831), OC43 (NC_006213), MERS (NC_019843), SARS-CoV-1 (NC_004718) and SARS-CoV-2 (MN908947.3) were downloaded from the NCBI database (https://www.ncbi.nlm.nih.gov/).

# Field-specific reporting

Please select the one below that is the best fit for your research. If you are not sure, read the appropriate sections before making your selection.

☒ Life sciences ☐ Behavioural & social sciences ☐ Ecological, evolutionary & environmental sciences

For a reference copy of the document with all sections, see nature.com/documents/nr-reporting-summary-flat.pdf

# Life sciences study design

All studies must disclose on these points even when the disclosure is negative.

| | |
|---|---|
| Sample size | Patients were recruited and patient material was banked at the University Hospital Freiburg; inclusion criteria were: (1) 32 health care workers that received a prime and boost vaccination with the mRNA vaccine bnt162b2/Comirnaty, (2) 59 acutely infected and convalescent individuals following a mild course of SARS-CoV-2 infection, SARS-CoV-2 infection was confirmed by positive PCR testing from oropharyngeal swab and/or SARS-CoV-2 spike IgG positive antibody testing, (3) 2 convalescent health care workers following a mild course of SARS-CoV-2 infection that received a single dose of bnt162b2/Comirnaty and (4) 8 age and sex-matched historic controls. No sample size calculations were performed. 32 vaccinatedhealth care workers gave informed consent and were available to donate blood samples. Therefore, similar numbers of COVID-19 convalescents were selected. |
| Data exclusions | For flow cytometrical analysis, cell populations containing less than 5 cells were excluded. This data exclusion strategy has been applied and validated previously by our group to gain reproducible results in studies investigating virus-specific CD8+ T cells in human viral infections. |
| Replication | Analyses were performed in independent experiments. Findings were reproducible. Flow cytometry analysis: 5 longitudinally analyzed vaccinees for A*01/S865 (n= 5 at BL, 4 at 3-4 dpp, 5 at 6-8 dpp, 5 at 9-12 dpp, 5 at 13-15 dpp, 2 at 16-18 dpp, 1 at 19-21 dpp, 4 at 3-4 dpb, 3 at 5-6 dpb, 2 at 7-9 dpb, 4 at 13-16 dpb, 4 at 28-30 dpb, 4 at 55-58 dpb and 5 at 80-120 dpb), 5 longitudinally anayzed vaccinees for A*02/S269 (n=5 at BL, 3 at 3-4 dpp, 4 at 6-8 dpp, 4 at 9-12 dpp, 5 at 13-15 dpp, 3 at 16-18 dpp, 2 at 19-21 dpp, 5 at 3-4 dpb, 2 at 5-6 dpb, 1 at 7-9 dpb, 5 at 13-16 dpb, 5 at 28-30 dpb, 4 at 55-58 dpb and 5 at 80-120 dpb) and 4 longitudinally analyzed vaccinees for A*03/S378 (n= 4 at BL, 3 at 3-4 dpp, 4 at 6-8 dpp, 4 at 9-12 dpp, 4 at 13-15 dpp, 1 at 19-21 dpp, 2 at 3-4 dpb, 2 at 5-6 dpb, 3 at 7-9 dpb, 3 at 13-16 dpb, 3 at 28-30 dpb and 4 at 80-120 dpb) in independent experiments (Figure 1, 2, Extended Data Figure 2d, e, 3, 4d, e, f, 5c), 2 longitudinally analyzed convalescent health care workers following a mild course of SARS-CoV-2 infection that received a single dose of bnt162b2/Comirnaty for A*02/S269 (n=2 at convalescent time point prior vaccination, n=2 at 3-4 dpp, n=2 at 8-9 dpp, n=2 at 13-15 dpp and n=2 at 22-23 dpp) in independent experiments (Extended Data Figure 4a, b, c), 8 longitudinally analyzed vaccinees for DRB1*15:01/S236 (n= 8 at BL, 5 at 3-4 dpp, 6 at 6-8 dpp, 7 at 9-12 pp, 6 at 13-15 dpp, 4 at 16-18 dpp, 6 at 3-4 dpb, 4 at 5-6 dpb, 3 at 7-9 dpb, 4 at 13-16 dpb, 7 at 28-30 dpb, 1 at 55-58 dpb and 1 at 84-86 dpb) in independent experiments (Figure 3a, b, c, Extended Data Figure 5e, f, g), 8 cross-sectionally analyzed historic controls (DRB1*15:01/S236) in independent experiments (Extended Data Figure 6d), 8 longitudinally analyzed vaccinees for S1 (n=8 at BL, 8 at 13-15 dpp, 2 at16-18 dpp, 3 at 19-21 dpp, 5 at 2-4 dpp, 3 at 5-6 dpp, 7 at 28-30 dpp) and RBD (n=8 at BL, 8 at 13-15 dpp, 2 at16-18 dpp, 3 at 19-21 dpp, 5 at 2-4 dpp, 3 at 5-6 dpp, 7 at 28-30 dpp), in independent experiments (Figure 3b, f, Extended Data Figure 7c), and 10 donors with a history of natural SARS-CoV-2 infection cross-sectionally for S1 and 10 cross-sectionally for RBD in independent experiments (Figure 3f, g, Extended Data Figure 7c), 5 longitudinally analyzed vaccinees for A*01/S865 (n=5 at 20-40dpb and 4 at 40-80 dpb), 5 longitudinally analyzed vaccinees for A*02/S269 (n=5 at 20-40dpb and 4 at 40-80 dpb), 4 longitudinally analyzed vaccinees for A*03/S378 (n=4 at 20-40dpb and 2 at 40-80 dpb), 9 donors with a history of natural SARS-CoV-2 infection cross-sectionally for A*01/S865 (n=5 at 20-40dpb, 4 at 40-80 dpb), 8 donors with a history of natural SARS-CoV-2 infection cross-sectionally for A*02/S269 (n=4 at 20-40dpb, 4 at 40-80 dpb), 7 donors with a history of natural SARS-CoV-2 infection cross-sectionally for A*03/S378 (n=4 at 20-40dpb, 3 at 40-80 dpb) in independent experiments (Extended Data Figure 8a, b, 9b, c), 11 cross-sectionally analyzed vaccinees for A*01/S865 at 80-120 dpb, 9 cross-sectionally anayzed vaccinees for A*02/S269 at 80-120 dpb, 8 cross-sectionally anayzed vaccinees for A*03/S378 at 80-120 dpb, 10 donors with a history of natural SARS-CoV-2 infection cross-sectionally for A*01/S865 80-120 dpb, 10 donors with a history of natural SARS-CoV-2 infection cross-sectionally for A*02/S269 80-120 dpb and 10 donors with a history of natural SARS-CoV-2 infection cross-sectionally for A*03/S378 80-120 dpb in independent experiments (Figure 4a, b, c, d, Extended Data Figure 9c, 10). ELISA and NT analysis: 8 longitudinally analyzed vaccinees for anti-SARS-CoV-2-S1 IgG (n= 8 at BL, 5 at 3-4 dpp, 7 at 6-8 dpp, 7 at 9-12 dpp, 8 at 13-15 dpp, 4 at 16-18 dpp, 3 at 19-21 dpp, 7 at 3-4 dpb, 4 at 5-6 dpp, 2 at 7-9 dpp, 7 at 13-16 dpp, 7 at 28-30 dpp, 8 at 55-58 dpb and 4 at 84-86 dpb) in independent experiments (Figure 3d), 8 longitudinally analyzed vaccinees for B.1 neutralizing titer (n= 6 at BL, 5 at 3-4 dpp, 6 at 6-8 dpp, 7 at 13-15 dpp, 2 at 19-21 dpp, 6 at 3-4 dpb, 3 at 5-6 dpp, 7 at 13-16 dpp, 7 at 28-30 dpp, 7 at 55-58 dpb and 5 at 84-86 dpb) in independent experiments (Figure 3d, Extended Data Figure 3e), 7 longitudinally analyzed vaccinees for alpha neutralizing titer (n=6 at BL, 6 at 6-8 dpp, 7 at 13-16 dpp, 2 at 19-21 dpp, 6 at 3-4 dpb, 3 at 5-6 dpp, 7 at 13-16 dpp, 5 at 28-30 dpp, 7 at 55-58 dpb and 5 at 84-86 dpp) and 7 longitudinally analyzed vaccinees for beta neutralizing titer |

(n=6 at BL, 6 at 6-8 dpp, 7 at 13-16 dpb, 2 at 19-21 dpp, 6 at 3-4 dpb, 3 at 5-6 dpb, 7 at 13-16 dpb, 5 at 28-30 dpb, 7 at 55-58 dpb and 5 at 84-86 dpb) in independent experiments (Extended Data Figure 3d), 16 donors with a history of natural SARS-CoV-2 infection cross-sectionally/longitudinally for B.1 neutralizing titer (n=4 at 20-40 dps, 5 at 40-80 dps and 7 at >80 dps) in independent experiments (Extended Data Figure 7e), 9 longitudinally analyzed vaccinees for anti-SARS-CoV-2 S1 IgM supernatant (n=9 at BL, 9 at 13-15 dpp, 2 at 16-18 dpp, 3 at 19-21 dpp, 5 at 2-4 dpb, 4 at 5-6 dpb and 9 at 28-30 dpb) and 4 donors with a history of natural SARS-CoV-2 infection cross-sectionally for anti-SARS-CoV-2 S1 IgM supernatant (n=9 at BL, 9 at 13-15 dpp, 2 at 16-18 dpp, 3 at 19-21 dpp, 5 at 2-4 dpb, 4 at 5-6 dpb and 9 at 28-30 dpb) and 8 longitudinally analyzed vaccinees for S1 binding IgG production after re-stimulation (n=8 at BL, 8 at 13-15 dpp, 2 at 16-18 dpp, 3 at 19-21 dpp, 5 at 2-4 dpb, 3 at 5-6 dpb and 7 at 28-30 dpb) and 8 donors with a history of natural SARS-CoV-2 infection cross-sectionally in independent experiments (Figure 3g). Overlapping peptides analyses: 16 analyzed vaccinees for 182 overlapping peptides spanning the SARS-CoV-2 Spike sequence (9-59 dpb) in independent experiments (Extended Data Figure 1d).

| Randomization | Vaccinated donors and donors with a history of natural SARS-CoV-2 infection were selected based on availability and HLA-typing. To analyze A*01/S865, A*02/S269, A*03/S378 and DRB1*15:01/S236-specific T cells participants needed to be allocated into experimental groups based on their HLA-typing. The covariates age and gender are well-documented: Median age of vaccinated donors was 39,6 years, donors with a history of natural SARS-CoV-2 infection was 47,2 years, donors with a history of natural SARS-CoV-2 vaccination and a single vaccination was 56,5 years, of historic controls 37,6 years. The gender ratio of vaccinated donors was m/f: 19/13, donors with a history of natural SARS-CoV-2 infection was m/f: 31/28, donors with a history of natural SARS-CoV-2 vaccination and a single vaccination was m/f: 1/1, of historic controls m/f: 5/3. |
|---|---|
| Blinding | Blinding was not applied. Non-objective parameters were not included in the study design. Due to standardized analyses of the flow cytometric data set, biased analysis can be excluded. |

# Reporting for specific materials, systems and methods

We require information from authors about some types of materials, experimental systems and methods used in many studies. Here, indicate whether each material, system or method listed is relevant to your study. If you are not sure if a list item applies to your research, read the appropriate section before selecting a response.

### Materials & experimental systems

| n/a | Involved in the study |
|---|---|
| ☐ | ☒ Antibodies |
| ☒ | ☐ Eukaryotic cell lines |
| ☒ | ☐ Palaeontology and archaeology |
| ☒ | ☐ Animals and other organisms |
| ☐ | ☒ Human research participants |
| ☒ | ☐ Clinical data |
| ☒ | ☐ Dual use research of concern |

### Methods

| n/a | Involved in the study |
|---|---|
| ☒ | ☐ ChIP-seq |
| ☐ | ☒ Flow cytometry |
| ☒ | ☐ MRI-based neuroimaging |

## Antibodies

| Antibodies used | T cell analysis |
|---|---|
| | BD Biosciences: |
| | anti-CCR7-PE-CF594 (150503, 1:50), Cat# 353232 |
| | anti-CCR7-BUV395 (3D12, 1:25), Cat# 740267 |
| | anti-CD4-BV786 (L200, 1:200), Cat# 563914 |
| | anti-CD8-BUV395 (RPA-T8, 1:400), Cat# 563795 |
| | anti-CD8-BUV510 (SK1, 1:100), Cat# 563914 |
| | anti-CD8-APC (SK-1, 1:200), Cat# 345775 |
| | anti-CD11a-BV510 (HI111, 1:25), Cat# 563480 |
| | anti-CD28-BV421 (CD28.2, 1:100), Cat# 562613 |
| | anti-CD28 pure (CD28.2, 1:1000), Cat# 555726 |
| | anti-CD38-APC-R700 (HIT2, 1:400), Cat# 564980 |
| | anti-CD38-BUV737 (HB7, 1:200), Cat# 564686 |
| | anti-CD39-BV650 (TU66, 33:1), Cat# 563681 |
| | anti-CD45RA-BUV496 (HI100, 1:800), Cat# 750258 |
| | anti-CD45RA-BUV737 (HI100, 1:200), Cat# 564442 |
| | anti-CD69-BUV395 (FN50, 1:50), Cat# 564364 |
| | anti-CD107a-APC (H4A3, 1:100), Cat# 560664 |
| | anti-CD127-BUV737 (HIL-7R-M21, 1:50), Cat# 612795 |
| | anti-CD127-BV421 (HIL-7R-M21, 3:100), Cat# 562436 |
| | anti-Granzyme B-PE-CF594 (GB11, 1:100), Cat# 562462 |
| | anti-ICOS-BV711 (DX29, 1:100), Cat# 563833 |
| | anti-IFN-γ-FITC (25723.11, 1:8), Cat# 340449 |
| | anti-IL-21-PE (3A3-N2.1, 1:25), Cat# 560463 |
| | anti-PD-1-BV605 (EH12.1, 1:50), Cat# 563245 |
| | anti-PD-1-PE-Cy7 (EH12.2H7, 1:200), Cat# 561272 |
| | anti-PD-1-BV786 (EH12.1, 1013122, 3:100), Cat# 563789 |
| | anti-T-BET-PE-CF594 (O4-46,93533305, 3:100), Cat# 562467 |
| | anti-TNF-PE-Cy7 (Mab11, 1:400), Cat# 557647 |
| | ViaProbe (7-AAD, 1:33), Cat# 555816 |

BioLegend
anti-BCL-2-BV421 (100, 1:200), Cat# 658709
anti-CCR7-BV785 (G043H7, 1:50), Cat# 353230
anti-CD4-AlexaFluor700 (RPA-T4, 300526, 1:200), Cat# 300526
anti-CD25-BV650 (BC96, 1:33), Cat# 302633
anti-CD57-BV605 (QA17A04, 1:100), Cat# 563895
anti-CD127-BV605 (A019D5, 3:100), Cat# 351334
anti-CXCR3-PerCP-Cy5.5 (G025H7, 1:33), Cat# 353714
anti-CXCR3-BV510 (G025H7, 3:100), Cat# 353726
anti-CXCR5-BV421 (J252D4, 1:100), Cat# 356920
anti-IL-2-PerCP-Cy5.5 (MQ1-17H12, 1:100), Cat# 500322
anti-Ki67-BV711 (Ki-67, 1:200), Cat# 350516
anti-Ki67-PE-Cy7 (Ki67, 1:200), Cat# 350504

Cell Signaling,
anti-TCF1-AlexaFluor488 (C63D9, 1:100), Cat# 6444

eBioscience
anti-CD14-APC-eFluor780 (61D3, 1:400), Cat#  47-0149-42
anti-CD19-APC-eFluor780 (HIB19, 1:400), Cat#  47-0199
anti-CD27-FITC (0323, 1:100), Cat# 11-0279
anti-KLRG1-BV711 (13F12F2, 1:50), Cat#  67-9488-42
anti-T-BET-PE-Cy7 (4B10, 1:200), Cat#  25-5825
anti-TOX-eFluor660 (TRX10, 1:100),  Cat# 50-6502
anti-EOMES-PerCP-eF710 (WD1928, 1:50), Cat# 46-4877-42
Viability Dye (APC-eFluor780 1:200, 1:400) Cat#  65-0865

Invitrogen

anti-CD45RA-PerCP-Cy5.5 (HI100, 3:100), Cat# 45-0458-42

B cell analysis

BioLegend
anti-CD20-BV510 (2H7, 1:80), Cat# 302340
anti-IgM-BV605 (MHM-88, 1:200), Cat# 314524
anti-CD24-FITC (ML5, 1:1000), Cat# 334103
anti-CD71-FITC (CY1G4, 1:1000), Cat# 334103
anti-CD95-PE-Dazzle594 (DX2, 1:50), Cat# 305634
anti-CD38-PE-Cy7 (HB-7), Cat# 356608
anti-BAFF-R-AF647 (11C1, 1:100), Cat# 316914
anti-CD19-APC-Cy7 (HIB19, 1:150), Cat# 302218
Zombie NIR Fixable Viability Kit (1:800), Cat# 423106

BD Biosciences
anti-IgG-BV650 (G18-145, 1:600), Cat# 740596
anti-CD27-BV786 (L128, 1:100), Cat# 563327
anti-CD69-BV480 (FN50, 1:200), Cat# 747519

Jackson ImmunoResearch
anti-IgA-PerCP (polyclonal, 1:200), Cat# 109-125-011

Invitrogen
anti-CD3-SB436 (OKT3, 1:200), Cat# 62-0037-42
anti-CD33-SB436 (WM-53, 1:50) Cat# 62-0338-42
anti-IgD-PerCP-eFluor 710 (IA6-2, 1:200), Cat# 46-9868-42

| Validation | All antibodies were obtained from commercial cendors and we based specificity on descriptions and information provided in corresponding data sheets available and provided by the manufacturers. Standardized analysis in different cohorts, antibody titration on PBMCs including unstained controls, comparisons of different antibody clones and conjugates and validated by publications: CCR7, clone 3D12 and 150503: antibody titration on PBMCs; control clone G043H7; validated with respect to differential expression of naïve and non-naïve T cell subpopulations CD4, clone L200 and RPA-T4: antibody titration on PBMCs; control clones SK3; using B cells as negative control CD8, clone SK1 and RPA-T8: antibody titration on PBMCs; control clone GHI/75; using B cells as negative control CD11, clone HI111: antibody titration on PBMCs, control clones TS2/4 and G43-25B; validated with respect to differential expression of activated and non-activated T cell subpopulations CD27, clone O323: antibody titration on PBMCs; control clone L128; validated with respect to differential expression of naïve and non-naïve T cell subpopulations CD28,clone CD28.2: antibody titration on PBMCs; control clone B-T3; validated with respect to differential expression of naïve and non-naïve T cell subpopulations CD45RA, clone HI100: antibody titration on PBMCs; validated with respect to differential expression of naïve and non-naïve T cell subpopulations CD69, clone FN50: antibody titration on PBMCs; validated with respect to differential expression of activated and non-activated T cell subpopulations CD107a, clone H4A3: antibody titration on PBMCs; validated with respect to differential expression of activated and non-activated T |
|---|---|

cell subpopulations

CD127, clone HIL-7R-M21 and A019D5: antibody titration on PBMCs; control clone eBioRDR5; validated with respect to differential expression of naïve and non-naïve T cell subpopulations

Granzyme B, clone GB11: antibody titration on PBMCs; polyclonal antibody as control; validated with respect to differential expression of activated and non-activated T cell subpopulations

IFNγ, clone 25723.11: antibody titration on PBMCs; control clone 4S.B3; validated with respect to differential expression of activated and non-activated T cell subpopulations

IL-21, clone 3A3-N2.1: antibody titration on PBMCs; validated with respect to differential expression of activated and non-activated T cell subpopulations

PD-1, clone EH12.2H7 and EH12.1:  antibody titration on PBMCs; control clone eBioJ105; validated with respect to differential expression of naïve and non-naïve T cell subpopulations

TNF, clone MAb11: antibody titration on PBMCs; validated with respect to differential expression of activated and non-activated T cell subpopulations

Bcl-2 , clone 100: antibody titration on PBMCs; validated with respect to differential expression of naïve  and non-naïve T cell subpopulations

CD25, clone BC96: antibody titration on PBMCs; control clone M-A251; validated with respect to differential expression of activated and non-activated T cell subpopulations

CD38, clone HB7 and HIT2: antibody titration on PBMCs; control clone HIT2.1; validated with respect to differential expression of naïve and non-naïve T cell subpopulations

CD39, clone TU66: antibody titration on PBMCs; control clones eBioA1 and A1; validated with respect to differential expression of naïve and non-naïve T cell subpopulations

CD57, clone QA17A04: antibody titration on PBMCs; control clone NK-1; validated with respect to differential expression of naïve and non-naïve T cell subpopulations

CXCR3, clone G025H7: antibody titration on PBMCs; control clone 1C6/CXCR3; validated with respect to differential expression of activated and non-activated T cell subpopulations

IL-2, clone MQ1-17H12: antibody titration on PBMCs; validated with respect to differential expression of activated and non-activated T cell subpopulations

TCF-1, clone C63D9: antibody titration on PBMCs; control clone 7F11A10; validated with respect to differential expression of naïve and non-naïve T cell subpopulations

Eomes, clone WD1928: antibody titration on PBMCs; validated with respect to differential expression of naïve and non-naïve T cell subpopulations

CD14 , clone 61D3: antibody titration on PBMCs; control clones M5E2 and   MφP9; using T cell populations as negative control

CD19, clone HIB19: antibody titration on PBMCs; control clone SJ25C1; using T cell populations as negative control

T-bet, clone O4-46: antibody titration on PBMCs; control clones 4B10; validated with respect to differential expression of naïve and non-naïve T cell subpopulations

KLRG1, clone 13F12F2: antibody titration on PBMCs; validated with respect to differential expression of naïve and non-naïve T cell subpopulations

TOX1, clone TRX10: antibody titration on PBMCs; control clone REA473; validated with respect to differential expression of naïve and non-naïve T cell subpopulations

CXCR5, clone J252D4: antibody titration on PBMCs; control clone MU5UBEE; validated with respect to differential expression of naïve and non-naïve T cell subpopulations

ICOS, clone DX29: antibody titration on PBMCs; control clone ISA-3; validated with respect to differential expression of naïve and non-naïve T cell subpopulations

Ki67, clone Ki67: antibody titration on PBMCs; control clone B56; validated with respect to differential expression of naïve and non-naïve T cell subpopulations

Viability Dye was titrated on PBMCs; validated with respect to differential staining of live and dead cell populations

CD20, clone 2H7: Titration on fresh and cryopreserved PBMCs, staining compared to clone JDC-10 and polyclonal anti-IgM Ab (Jackson ImmunoResearch)

IgM clone MHM-88: Titration on fresh and cryopreserved PBMCs, staining compared to clone JDC-10 and polyclonal anti-IgM Ab

CD71 clone CY1G4: Titration on fresh PBMCs and in vitro activated B cells

CD95 clone DX2: Titration on fresh PBMCs and in vitro activated B cells

CD38 clone HB-7: Titration on fresh and cryopreserved PBMCs; staining compared to clone HIT2

BAFF-R clone 11C1: Titration on cryopreserved PBMCs

CD19 clone HIB19: Titration on cryopreserved PBMCs

IgG clone G18-145: Titration on fresh and cryopreserved PBMCs, staining compared to clone JDC-10 and polyclonal anti-IgG Ab

CD27 clone L128: Titration on fresh and crypreserved PBMCs, staining compared to clone LG.3A10

IgA (polyclonal): Titration on fresh and cryopreserved PBMCs

IgD clone IA6-2: Titration on fresh and cryopreserved PBMCs

CD24,  clone ML5: Titration on fresh PBMCs and in vitro activated B cells

CD3, clone OKT3:Titration on cryopreserved PBMCs

CD33, clone WM-53: Titration on cryopreserved PBMCs

CD69, clone FN50: Titration on in vitro activated B cells

# Human research participants

Policy information about studies involving human research participants

| Population characteristics | 32 health care workers that received a prime and boost vaccination with the mRNA vaccine bnt162b2/Comirnaty, 59 acutely infected and convalescent individuals following a mild course of SARS-CoV-2 infection, 2 convalescent health care workers following a mild course of SARS-CoV-2 infection that received a single dose of bnt162b2/Comirnaty and 8 historic controls, carrying either of the following HLA alleles: A*01:01, -A*02:01, -A*03:01,  DRB1*15:01 were recruited at the University Hospital Freiburg. Median age of vaccinated donors was 39,6 years, donors with a history of natural SARS-CoV-2 infection was 47,2 years,  donors with a history of natural SARS-CoV-2 vaccination and a single vaccination was 56.5 years, of historic |

controls 37,6 years. The gender ratio of vaccinated donors was m/f: 19/13, donors with a history of natural SARS-CoV-2 infection was m/f: 31/28, donors with a history of natural SARS-CoV-2 vaccination and a single vaccination was m/f: 1/1, of historic controls m/f: 5/3.

| | |
|---|---|
| Recruitment | Vaccinated donors as well as SARS-CoV-2-infected and SARS-CoV-2-convalescent patients were recruited at the University Hospital Freiburg (in- and outpatient section); self-selection bias or other biases can be excluded since several people were included in the recruitment. Samples were banked and retrospectively selected according to the following inclusion criteria: HLA-A*01:01, -A*02:01, -A*03:01, DRB1*15:01. Banked samples from sex-, age- and HLA-matched historic controls were retrospectively selected. |
| Ethics oversight | Written informed consent was obtained from all participants and the study was conducted according to federal guidelines, local ethics committee regulations (Albert-Ludwigs-Universität, Freiburg, Germany; vote #: 322/20, #21-1135 and 315/20) and the Declaration of Helsinki (1975). |

Note that full information on the approval of the study protocol must also be provided in the manuscript.

# Flow Cytometry

## Plots

Confirm that:

☒ The axis labels state the marker and fluorochrome used (e.g. CD4-FITC).

☒ The axis scales are clearly visible. Include numbers along axes only for bottom left plot of group (a 'group' is an analysis of identical markers).

☒ All plots are contour plots with outliers or pseudocolor plots.

☒ A numerical value for number of cells or percentage (with statistics) is provided.

## Methodology

| | |
|---|---|
| Sample preparation | Cryopreserved isolated human PBMCs were thawed and prepared for flow cytometry or in vitro expansion described in the methods section |
| Instrument | FACSCanto II, LSRFortessa (BD, Germany), Cytek Aurora (Cytek) or CytoFLEX (Beckman Coulter), Tecan (LifeScience) |
| Software | FlowJo_v10.6.2 (Treestar), R version 4.0.2 using the Bioconductor (version: Release (3.11)) . |
| Cell population abundance | Abundance of SARS-CoV-2-specific T cells are low (<10^-4 %) |
| Gating strategy | CD8+ T cells: Lymphocytes gated on FSC-A and SSC-A, Doublet exclusion on FSC-A and FSC-H and FSC-A and FSC-W, Exclusion of dead cells, B cells and monocytes, Gating on CD8+ cells, Exclusion of naive cells (CCR7+CD45RA+), Gating of SARS-CoV-2-specific CD8+ T cells via tetramers described in methods part. |

☒ Tick this box to confirm that a figure exemplifying the gating strategy is provided in the Supplementary Information.

