## [Peer Review File · Nature]

Manuscript Title: Rapid and stable mobilization of CD8+ T cells by SARS-CoV-2 mRNA vaccine

Editorial Notes:

Reviewer Comments & Author Rebuttals

Reviewer Reports on the Initial Version:

Referee #1 (Remarks to the Author):

The manuscript "Rapid and stable mobilization of fully functional spike-specific CD8+ T cells preceding a mature humoral response after SARS-CoV-2 mRNA vaccination" reports a detailed characterization of the induction and persistence (until day 80 after second dose) of cellular components of adaptive immunity (CD 8 and CD4 T cells) induced by vaccination with mRNA vaccines in healthy, not previously SARS-CoV2 infected, individuals (total 17 individuals).

The authors performed a comprehensive longitudinal study (samples taken at multiple time points before and after second dose) of CD8 T cell populations specific for two Spike epitopes and of a single CD4 T cell epitope utilizing HLA-class I and HLA-class II tetramers. They reported that CD8 T cells are mobilized early and present a dynamic phenotypic profile. CD4 T cells and antibodies appear later (mainly after the second dose). The authors also claim that vaccination and natural infection induced "a different subset distribution dominated by effector memory T cells at the expense of self-renewing and multipotent central memory T cells" in vaccinated individuals.

The work is technically very well executed.

The finding that a certain quantity of Spike-specific T cells are present at early time points after the first dose vaccination, before the appearance of antibodies, is important because it suggests a peculiar role of cellular immunity in the protection against SARS-CoV-2-mediated disease. However, such observation is not completely novel since similar findings have been already recently published (Kalimuddin, S., Tham, C.Y., Qui, M., de Alwis, R., et al, Early T cell and binding antibody responses are associated with Covid-19 RNA vaccine efficacy onset, Med (2021), doi:

<https://doi.org/10.1016/j.medj.2021.04.003>).

The observation that Spike-specific CD8 T cells induced by vaccination present a distinct phenotype (more central memory) from Spike-specific CD8 T cells induced by natural infection is novel and potentially important. It might suggest a reduced ability of mRNA vaccines to induce a long-term memory T cell response. However, this analysis was performed only in a limited number of individuals (4 versus 3) and it is robustly supported only by results obtained in Spike-T cells specific for the A1-restricted epitope (Figure 4B).

Major comments:

a) The authors claim that CD8 T cells induced by vaccines can be the key of protection. The frequencies of the tetramer + CD8 T cells is however extremely low (see results without enrichment - extended Figure 2). This won't be a concern (at least to this reviewer) if the tetramer-analysed CD8 T cells would be representative of a much broader multi-epitope Spike-specific CD8 T cell response. It will be extremely informative, in my opinion, if at least some vaccinated individuals would be analyzed for total T cell responses against Spike using for example peptides covering all the Spike protein to understand whether the characterized CD8 T cells are part of a much more robust CD8 T cell response.

b) The dissimilar Spike-specific CD8 T cell phenotype reported in individuals 80 days after vaccination versus COVID-19 convalescents is an important observation. However, not only many more individuals need to be tested, but it is also not clear what means > 80 days in convalescents. The authors will need to describe better whether the time after infection or recovery in convalescent is

similar to the time after the second dose of the vaccinated. In addition, it will be very important to extend the analysis to other epitopes since the results observed in the two different epitopes are not identical.

Referee #2 (Remarks to the Author):

The manuscript describes a cohort of healthcare workers frequently sampled following mRNA vaccination for SARS-CoV-2. The major focus is on the phenotype and kinetic response of two Class I restricted Spike epitopes. There is also an analysis of a single Class II epitope, and a comparison to the induction of antibody and antigen-specific B cell responses. The authors conclude that: 1) CD8 T cell responses are induced more rapidly than other arms of the immune response; 2) Memory T cell differentiation is greater after infection than vaccination; 3) Response magnitudes for CD8 T cells are similar between vaccination and infection.

The major strength of the paper is the frequency of sampling through the entire vaccination period. This is very useful and shows relatively similar dynamics between the two measured epitopes, particularly for some phenotypic features.

The data are only from two epitopes and there is some degree of variability between them, which makes it difficult to project how these data relate to the overall response. Additionally there are no "recall" responses here which we know occur after infection and vaccination, recruiting cross-reactive specificities from prior coronavirus infections. The interpretation of the data is exaggerated in several places in the paper as detailed below.

1) A major point of the paper is that vaccination does not induce as robust "central memory" phenotypes as infection. This is a potentially controversial claim and should be strongly grounded by the data. The data in Figure 4B show a fair degree of noise, though the T-CM subset does appear entirely absent in the A01 epitope after d80. What is confusing is that in figure 1, the same epitope appears to have no change in TCF1+ CD8+ T cells across this time period or in CD127+ cells. The data in extended figure 7 also support this. I was unable to figure out how they were defining TCM in the methods, main text, or figure legend, which should be corrected regardless, but I'm not particularly confident in the significance of this claim and it is central to the current manuscript.

2) The authors argue that the CD4 and humoral responses are delayed relative to the CD8 response, but I don't think this is supported by the data either. They do acknowledge that much of the activity for humoral immunity occurs out of the blood, but they somewhat down play the fact that there is robust induction of IgG and neutralization activity in the d13-15 group. This is not so different for the majority of individuals compared to the CD8 response.

3) Related to 1 and 2, there are several claims in the summary that seem unsupported statistically to me. Line 53-54—a stable memory precursor pool of spike specific CD* T cells and fully functional spike specific effector CD8 T cells populations are vigorously mobilized as early as one week after prime vaccination, when CD4 T cell and spie-specific antibody responses are still weak and neutralizing antibodies are lacking". I can't tell if they've compared the same number of people across all these measures, but there are certainly people with no detectable CD8s until d13 and there are those with detectable antibody boosts earlier. This should be statistically rigorously analyzed or couched in much less definitive terms.

4) Similarly, they make the claim on line 62 about vaccine induced memory having more effectors "at the expense of self-renewing and multipotent central memory t cells." Again, these data seem to contradict what is clear in Figure 1, vs. the undefined CM populations in Figure 4.

5) Line 63 they suggest that CD8s "may represent the major correlate of early protection"—this is completely unsupported and should be removed. Even couched with a "may" a correlate of protection is a technical claim that the authors have in no way established.

6) In the discussion, the authors claim that there is no effect of the boost on the CD8 compartment ("transient effects") but without a proper comparison to a group that only receives one vaccine vs. the prime-boost this claim should be removed. Where these data are described in the paper (lines 145-147) I think they are also overinterpreted—yes the frequencies are stable (despite later claims that there are no central memory cells later after the boost) throughout the prime to boost period,

but with this correlational analysis the authors can't conclude that there isn't some replacement into the memory pool by the boost response.

Referee #3 (Remarks to the Author):

Oberhard et al investigated spike-specific CD8+ T cells, CD4+ T cells, B cells and antibodies following SARS-CoV-2 mRNA vaccination. Using longitudinal PBMC samples from vaccinees after the first and second vaccine doses, the authors used tetramer enrichment protocols to define T cell responses at a single epitope level. They found rapid and persistent activation of spike-specific CD8+ T cells within a week after vaccination, when CD4+ T cell and B cell responses were relatively low. In comparison to SARS-CoV-2 infection-induced CD8+ T cells, tetramer-specific CD8+ T cell populations elicited by vaccination were mainly of an effector phenotype and fully functional. These findings suggest that spike-specific CD8+ T cells can represent a correlate of protection elicited by the mRNA vaccines. These results are of key importance for our understanding of immune responses elicited by COVID-19 vaccines. The data are of the highest quality, convincing and strongly support conclusions. The study is extremely well performed and presented, timely, important and interesting.

Minor comments:

Representative FACS plots should be shown for Fig 1D (CD38 vs tetramers; Ki-67 vs tetramers; T-BET vs tetramers); and Fig 1FG (CD127 and TCF-1 vs tetramers).

Authors analysed IFN-g, TNF and CD107a as hallmarks of CD8+ T cell functionality. Further analysis of the data in terms of polyfunctionality is recommended.

Throughout the manuscript, the statistical analyses are missing. Statistically significant results should be indicated with the asterisks on the figures.

Page 7: please rephrase "proper effector capacity" as it is unclear what the 'proper' effector capacity means.

Fig 3F: probe-specific B cell staining seems suboptimal. Have these PBMC samples been enriched for tetramer-specific B cells?

Discussion: the authors state that the lower CD38 expression level after vaccination (comparing to SARS-CoV-2 infection) might result from limited antigen recognition. This might also be, at least in part, driven by differences in inflammation during infection and vaccination.

Line 328: "only provide insides" should read "only provide insights"

Author Rebuttals to Initial Comments:

REFEREES' COMMENTS:

REFEREE #1:

The manuscript "Rapid and stable mobilization of fully functional spike-specific CD8+ T cells preceding a mature humoral response after SARS-CoV-2 mRNA vaccination" reports a detailed characterization of the induction and persistence (until day 80 after second dose) of cellular components of adaptive immunity (CD 8 and CD4 T cells) induced by vaccination with mRNA vaccines in healthy, not previously SARS-CoV2 infected, individuals (total 17 individuals).

The authors performed a comprehensive longitudinal study (samples taken at multiple time

points before and after second dose) of CD8 T cell populations specific for two Spike epitopes and of a single CD4 T cell epitope utilizing HLA-class I and HLA-class II tetramers. They reported that CD8 T cells are mobilized early and present a dynamic phenotypic profile. CD4 T cells and antibodies appear later (mainly after the second dose). The authors also claim that vaccination and natural infection induced “a different subset distribution dominated by effector memory T cells at the expense of self-renewing and multipotent central memory T cells” in vaccinated individuals.

The work is technically very well executed. The finding that a certain quantity of Spike-specific T cells are present at early time points after the first dose vaccination, before the appearance of antibodies, is important because it suggests a peculiar role of cellular immunity in the protection against SARS-CoV-2-mediated disease. However, such observation is not completely novel since similar findings have been already recently published (Kalimuddin, S., Tham, C.Y., Qui, M., de Alwis, R., et al, Early T cell and binding antibody responses are associated with Covid-19 RNA vaccine efficacy onset, *Med* (2021), doi: <https://doi.org/10.1016/j.medj.2021.04.003>). The observation that Spike-specific CD8 T cells induced by vaccination present a distinct phenotype (more central memory) from Spike-specific CD8 T cells induced by natural infection is novel and potentially important. It might suggest a reduced ability of mRNA vaccines to induce a long-term memory T cell response. However, this analysis was performed only in a limited number of individuals (4 versus 3) and it is robustly supported only by results obtained in Spike-T cells specific for the A1-restricted epitope (Figure 4B).

We would like to thank this reviewer for the positive feedback on our study emphasizing a potentially peculiar role of T cells in protecting from SARS-CoV-2-mediated diseases. As mentioned by this reviewer and also discussed in the introduction, Kalimuddin et al. have recently reported that T cells are detectable early after mRNA vaccination already pointing towards an important role of especially CD8+ T cells. The authors focused their analyses on the overall spike-reactive T cell response that is relevant to understand the breadth of the T cell response, however, underestimates the strength of the spike-specific CD8+ T cell response as recently reported by Sahin et al., *Nature* 2021. Furthermore, the analysis of spike-specific CD8+ T cell responses on a single epitope level enables to precisely assess the respective dynamics, trajectories and functional capacities and with this adds novel insights into the cellular immune response after mRNA vaccination.

Following this reviewer’s comment we have increased the numbers of individuals analyzed at later time-points (>day 80 post boost vaccination to n=28 and >day 80 post symptom onset to n=30, respectively; revised Fig. 4). A statistically significant increase in the proportion of central memory T cells has now been detectable for A*01/S₈₆₅ and A*02/S₂₆₉-specific CD8+ T cells after natural infection compared to vaccination (revised Fig. 4C), confirming our initial results.

Major comments:

a) The authors claim that CD8 T cells induced by vaccines can be the key of protection. The frequencies of the tetramer + CD8 T cells is however extremely low (see results without enrichment - extended Figure 2). This won’t be a concern (at least to this reviewer) if the tetramer-analysed CD8 T cells would be representative of a much broader multi-epitope Spike-specific CD8 T cell response. It will be extremely informative, in my opinion, if at least some vaccinated individuals would be analyzed for total T cell responses against Spike using for example peptides covering all the Spike protein to understand whether the characterized CD8 T cells are part of a much more robust CD8 T cell response.

We completely agree with this reviewer that it is important to relate the CD8+ T cell response targeting the epitopes analyzed here to the overall spike-specific CD8+ T cell response. To address this important point, we have analyzed spike-reactive CD8+ T cell responses of n=16 vaccinees after stimulation with overlapping peptides spanning the whole spike protein.

Importantly, the tested A*01/S₈₆₅- and A*02/S₂₆₉-specific CD8+ T cell responses as well as CD8+ T cell responses targeting the newly included epitope A*03/S₃₇₈ (see comment b of this reviewer) are part of a broader CD8+ T cell response targeting the spike protein with multiple epitopes restricted by the same HLA class I alleles as well as other frequently occurring HLA class I alleles (revised Extended Data Fig. 1C). Of note, however, the 3 epitopes selected by us proved to be the dominant responses in the background of the respective restricting HLA class I allele (revised Extended Data Fig. 1C). These important results are now described in the revised manuscript on page 5, lines 107-110.

b) The dissimilar Spike-specific CD8 T cell phenotype reported in individuals 80 days after vaccination versus COVID-19 convalescents is an important observation. However, not only many more individuals need to be tested, but it is also not clear what means > 80 days in convalescents. The authors will need to describe better whether the time after infection or recovery in convalescent is similar to the time after the second dose of the vaccinated. In addition, it will be very important to extend the analysis to other epitopes since the results observed in the two different epitopes are not identical.

According to this reviewer's comment, we have increased the numbers of individuals analyzed at >day 80 post boost vaccination to n=28 and >day 80 post symptom onset to n=30 (revised Fig. 4). In the revised Extended Data Figure 8A, we now depict a diagram showing the distribution of the analyzed time-points post boost vaccination and post symptom onset. The comparative analyses of CD8+ T cell responses after vaccination versus natural infection were performed in a time-point matched manner. We have better clarified this point in the revised manuscript (page 10, lines 239-242). In addition, we have also extended our analyses to CD8+ T cells targeting a third epitope (A*03/S₃₇₈). Indeed, all CD8+ T cells targeting the different spike epitopes (A*01/S₈₆₅, A*02/S₂₆₉ and A*03/S₃₇₈) differed in their phenotypic characteristics. We have now included this point in the revised results section depicting t-SNE analyses comparing the different CD8+ T cell responses in revised Figure 4B and revised Extended Data Figure 8C. For example, an increased proportion of A*01/S₈₆₅ and A*02/S₂₆₉-specific CD8+ T cells were within the central memory T cell compartment after natural infection compared to vaccination, but this was not the case for A*03/S₃₇₈-specific CD8+ T cells. These important findings are described on page 10, line 246, to page 11, line 264, of the revised manuscript.

REFEREE #2:

The manuscript describes a cohort of healthcare workers frequently sampled following mRNA vaccination for SARS-CoV-2. The major focus is on the phenotype and kinetic response of two Class I restricted Spike epitopes. There is also an analysis of a single Class II epitope, and a comparison to the induction of antibody and antigen-specific B cell responses. The authors conclude that: 1) CD8 T cell responses are induced more rapidly than other arms of the immune response; 2) Memory T cell differentiation is greater after infection than vaccination; 3) Response magnitudes for CD8 T cells are similar between vaccination and infection.

The major strength of the paper is the frequency of sampling through the entire vaccination period. This is very useful and shows relatively similar dynamics between the two measured epitopes, particularly for some phenotypic features.

The data are only from two epitopes and there is some degree of variability between them, which makes it difficult to project how these data relate to the overall response. Additionally, there are no "recall" responses here, which we know occur after infection and vaccination, recruiting cross-reactive specificities from prior coronavirus infections. The interpretation of the data is exaggerated in several places in the paper as detailed below.

We would like to thank this reviewer for the valuable comments that helped to specify our conclusions and thus to improve our manuscript. According to the comments of this reviewer and reviewer #1, we have (i) increased the sample size of the spike-specific CD8+ T cell analyses at later time-points to account for variability of vaccinees (see revised Figure 4); (ii) profiled CD8+ T cell responses targeting an additional epitope (A*03/S₃₇₈; see revised Figures 1, 2, 4 and revised Extended Data Figures 2, 3, 4, 5, 6, 8, 9, 10); and (iii) performed CD8+ T cell stimulation with overlapping peptides spanning the whole S protein to relate the in-depth analyzed spike-specific CD8+ T cell responses to the overall response (see Extended Data Figure 1C). In brief, deeply profiled A*01-, A*02- and A*03-restricted CD8+ T cell responses are part of a broader T cell response (revised Extended Data Fig. 1C), however, proved to be dominant when analyzing responses spanning the whole S protein (revised Extended Data Fig. 1C). Although several characteristics are shared between A*01/S₈₆₅-, A*02/S₂₆₉- and A*03/S₃₇₈-specific CD8+ T cell responses after vaccination (revised Figure 1), there are still phenotypic differences, especially in the early memory phase after vaccination and infection (revised Figure 4B and C and Extended Data Figure 8C).

Moreover, we agree with this reviewer that the boost response after second dose vaccination is not a “recall” response in the classical sense. We therefore rephrased the respective sentences avoiding the usage of “recall” (e.g., page 13, line 286; page 14, line 320 of the revised manuscript). In addition, we now also provide data analyzing spike-specific CD8+ T cells after first dose mRNA vaccination in 2 individuals who recovered from mild to moderate infection approx. 12 months ago resembling a more classical recall response (revised Extended Data Fig. 4A-C). In the two analyzed vaccinees with previous natural infection, a boost expansion of spike-specific CD8+ T cells was detectable accompanied by an increase of CD38, Ki67 and Tbet expression similar to vaccination of individuals without prior infection (revised Extended Data Figure 4 A, B; page 6, lines 139-143 of the revised manuscript). However, TOX expression was not as highly expressed in spike-specific CD8+ T cells at the peak of infection/vaccination compared to prime/boost vaccination indicating subtle differences in the respective immune responses (revised Extended Data Figure 4 C).

In addition, as suggested by the reviewer we have toned down and rephrased several interpretations as specified below to provide a more balanced description and discussion of our results in the revised manuscript.

1) A major point of the paper is that vaccination does not induce as robust “central memory” phenotypes as infection. This is a potentially controversial claim and should be strongly grounded by the data. The data in Figure 4B show a fair degree of noise, though the T-CM subset does appear entirely absent in the A01 epitope after d80. What is confusing is that in figure 1, the same epitope appears to have no change in TCF1+ CD8+ T cells across this time period or in CD127+ cells. The data in extended figure 7 also support this. I was unable to figure out how they were defining TCM in the methods, main text, or figure legend, which should be corrected regardless, but I’m not particularly confident in the significance of this claim and it is central to the current manuscript.

We thank this reviewer for this comment. In the revised version of the manuscript, we have included a more detailed gating strategy of T_{naïve}, T_{ED}, T_{CM}, T_{TM}, T_{EM1}, T_{EM2}, T_{EM3}, T_{EMRA} cells in Extended Data Figure 9A. As mentioned by the reviewer there are indeed inter-individual variations regarding memory T cell subset diversification. To further substantiate our conclusions, we increased the sample size of the phenotypic T cell analysis and additionally profiled CD8+ T cells targeting a third epitope (A*03/S₃₇₈). As depicted in the revised Figure 4C, our initial results were confirmed that the proportion of T_{CM} cells within A*01/S₈₆₅ and A*02/S₂₆₉-specific CD8+ T cells is higher following natural infection compared to vaccination. However, the proportion of T_{CM} cells within A*03/S₃₇₈-specific CD8+ T cells is similar after natural infection and vaccination. These data clearly highlight different characteristics of CD8+ T cell responses targeting different epitopes. We now have better clarified this point in the revised Figure 4B/C and in the revised manuscript on page 11, lines 250-264.

Since we stained CD127 together with other markers to discriminate the above-mentioned

memory T cell populations in the same panel, we were able to analyze CD127 expression of the distinct memory T cells subsets (Point-to-point-reply Fig. 1A). These analyses revealed that the discrepancy between the proportion of T_{CM} cells and CD127 expression probably results from the fact that CD127 expression is not limited to T_{CM} cells. In particular, about 66-100% of T_{ED}, 37-67% of T_{TM} and 25-53% of T_{EM1} expressed CD127. TCF1 was not included in the same panel with the markers to discriminate between all memory T cell subsets. However, we have analyzed TCF1 expression in T_{CM} and T_{EM} cells (Point-to-point-reply Fig. 1B). This analysis revealed that TCF1 expression was detectable in 64-84% of T_{CM} and 27-48% of T_{EM} cells.

Point-to-point-reply Fig 1: CD127 expression (A) and TCF1 expression (B) in distinct memory subsets, displayed separately for the three epitopes in vaccinees >80 dpb.

2) The authors argue that the CD4 and humoral responses are delayed relative to the CD8 response, but I don't think this is supported by the data either. They do acknowledge that much of the activity for humoral immunity occurs out of the blood, but they somewhat down play the fact that there is robust induction of IgG and neutralization activity in the d13-15 group. This is not so different for the majority of individuals compared to the CD8 response.

We thank the reviewer for pointing out this important point and agree that our results regarding the delay of humoral responses in comparison to CD8+ T cells need further discussion. As pointed out, we found peak responses of CD4+ T cells and the humoral response to be delayed compared to CD8+ T cells. In particular, neutralizing activity was hardly detectable at day 6-8 post prime vaccination, when CD8+ T cells were already detectable in a substantial proportion of vaccinees, and were still at the lower limit of detection at day 9-12 post prime vaccination (Fig. 3D, please regard the logarithmic scale). To better visualize this aspect, we have performed a direct comparison of the presence of spike-specific CD8+ T cells and neutralizing antibody activity at days 6-8 and 13-15 post prime vaccination, respectively, (Point-to-point

reply Fig. 2). This analysis shows a significant difference at d6-8 that lost significance at d13-15. Following this reviewer's comment, we have stated our own data more carefully and have discussed clearly that these data confirm recent data by other groups in the revised manuscript (page 3, lines 57 and 63-66; page 13, lines 284-285; page 13, lines 302-306 of the revised manuscript).

Point-to-point-reply Fig 2: A Proportion of tested vaccinees with epitope-specific CD8+ T cell responses and neutralizing antibodies (nAbs) at days 6-8 and 13-15, respectively. **B** Proportions of vaccinees that display a response with at least the half max. of their peak response (mostly present after boost). Two-sided Fisher's exact test was performed.

3) Related to 1 and 2, there are several claims in the summary that seem unsupported statistically to me. Line 53-54—a stable memory precursor pool of spike specific CD8 T cells and fully functional spike specific effector CD8 T cells populations are vigorously mobilized as early as one week after prime vaccination, when CD4 T cell and spike-specific antibody responses are still weak and neutralizing antibodies are lacking”. I can't tell if they've compared the same number of people across all these measures, but there are certainly people with no detectable CD8s until d13 and there are those with detectable antibody boosts earlier. This should be statistically rigorously analyzed or couched in much less definitive terms.

Indeed, the dense data points in Fig. 1C hinders the direct comparison of the early CD8+ T cell and antibody titers. Following the reviewer's very helpful suggestion, we now have directly stated the proportion of patients with a detectable CD8+ T cell response at day 6-8 post prime (page 5, lines 115-116 of the revised manuscript). A detailed comparison of CD8+ T cells and neutralizing activity at day 6-8 and 13-18 post prime vaccination is displayed in Point-to-point reply Fig. 2, clearly demonstrating a significant difference at d6-8 that lost significance at d13-15. Since we however also agree with the reviewer that these biological parameters show variation and are not completely clear-cut, we have also stated and discussed the data more carefully in the revised manuscript (page 3, lines 57 and 63-66; page 13, lines 284-285 and 302-306 of the revised manuscript).

4) Similarly, they make the claim on line 62 about vaccine induced memory having more effectors “at the expense of self-renewing and multipotent central memory t cells.” Again, these data seem to contradict what is clear in Figure 1, vs. the undefined CM populations in Figure 4.

As depicted in Point-to-point reply Fig. 1, neither CD127 nor TCF-1 is exclusively expressed by T_{CM} cells and also T_{EM} cells express these markers to a substantial fraction. These expression patterns probably account for our observation that we detect a minor fraction of T_{CM} cells while having nearly constant frequencies of CD127- and TCF-1-expressing spike-

specific CD8+ T cells. A detailed gating strategy of the memory T cell subsets is included in the revised Extended Data Figure 9A.

5) Line 63 they suggest that CD8s “may represent the major correlate of early protection”—this is completely unsupported and should be removed. Even couched with a “may” a correlate of protection is a technical claim that the authors have in no way established.

We agree with the reviewer that we indeed provide a temporal association, but not a proof of the protective effect of the early CD8+ T cell response. We have therefore removed the term “correlate of protection” and clarified this limitation in the abstract (page 3, line 63-66) as well as discussion (page 14, line 308) of the revised manuscript.

6) In the discussion, the authors claim that there is no effect of the boost on the CD8 compartment (“transient effects”) but without a proper comparison to a group that only receives one vaccine vs. the prime-boost this claim should be removed. Where these data are described in the paper (lines 145-147) I think they are also overinterpreted—yes the frequencies are stable (despite later claims that there are no central memory cells later after the boost) throughout the prime to boost period, but with this correlational analysis the authors can’t conclude that there isn’t some replacement into the memory pool by the boost response.

We agree with the reviewer that without a “prime only” control cohort, an important effect of the boost vaccination on vaccine-induced CD8+ T cells cannot be excluded. It was definitely not our intention to suggest that a boost vaccination is not needed to generate and maintain a stable vaccine-induced CD8+ T cell response. We have clarified this important unintended ambiguity in the revised manuscript (page 7, lines 153-154: “and being unaffected by boost vaccination” was removed; page 13, lines 300-301: “with only transient effects of boosting after 3 weeks” was removed).

REFEREE #3:

Oberhardt et al investigated spike-specific CD8+ T cells, CD4+ T cells, B cells and antibodies following SARS-CoV-2 mRNA vaccination. Using longitudinal PBMC samples from vaccinees after the first and second vaccine doses, the authors used tetramer enrichment protocols to define T cell responses at a single epitope level. They found rapid and persistent activation of spike-specific CD8+ T cells within a week after vaccination, when CD4+ T cell and B cell responses were relatively low. In comparison to SARS-CoV-2 infection-induced CD8+ T cells, tetramer-specific CD8+ T cell populations elicited by vaccination were mainly of an effector phenotype and fully functional. These findings suggest that spike-specific CD8+ T cells can represent a correlate of protection elicited by the mRNA vaccines. These results are of key importance for our understanding of immune responses elicited by COVID-19 vaccines. The data are of the highest quality, convincing and strongly support conclusions. The study is extremely well performed and presented, timely, important and interesting.

We would like to thank this reviewer for highlighting the strengths and relevance of our study.

Minor comments:

Representative FACS plots should be shown for Fig 1D (CD38 vs tetramers; Ki-67 vs tetramers; T-BET vs tetramers); and Fig 1FG (CD127 and TCF-1 vs tetramers).

According to this reviewer’s suggestion we changed the depiction of the representative FACS plots from histograms to dot plots showing tetramers versus the indicated marker molecules in the revised version of Figure 1F,G and Extended Data Figures 2B/D and 4F.

Authors analysed IFN-g, TNF and CD107a as hallmarks of CD8+ T cell functionality. Further analysis of the data in terms of polyfunctionality is recommended.

We thank this reviewer for this suggestion and included data analysis of polyfunctionality in the revised version of Figure 2G comparing the functionality of CD8+ T cells targeting A*01/S₈₆₅, A*02/S₂₆₉ and A*03/S₃₇₈ after prime and boost vaccination. In addition, we now also depict polyfunctionality analyses of spike-specific CD8+ T cells after natural infection versus vaccination in the revised Extended Data Figure 10B.

Throughout the manuscript, the statistical analyses are missing. Statistically significant results should be indicated with the asterisks on the figures.

According to this reviewer's comment, we have included statistical testing of the time-course data. Statistical tests used are stated in the respective figure legend and exact p-values are depicted for significant results. Exact p-values were chosen instead of asterisks according to the Nature guidelines. We also included the statement "All statistically significant results are marked with the respective exact p-value" in the revised figure legends.

Page 7: please rephrase "proper effector capacity" as it is unclear what the 'proper' effector capacity means.

We thank this reviewer for bringing up this point. For clarification, we have rephrased "proper" to "reasonable" in the revised manuscript (page 7, line 158, and page 8, line 173 of the revised manuscript).

Fig 3F: Probe-specific B cell staining seems suboptimal. Have these PBMC samples been enriched for tetramer-specific B cells?

The antigen-specific B cells have not been enriched from PBMC samples. We have now clearly labelled the data being not enriched in the revised Figure 3G ("ex vivo spike-specific B cells (not enriched)").

To validate our data, we have enriched S1-specific B cells from PBMC samples of 2 different donors by applying anti-PE microbeads (Miltenyi Biotech, Germany) (Point-to-point-reply Fig 3A). Similar to the data obtained from ex vivo detectable S1-specific B cells, enriched S1-specific B cells showed a progressive development of class switched IgG CD27+ memory cells in vaccinated individuals and thus confirmed the results obtained with ex vivo detectable antigen-specific B cells (Point-to-point-reply Fig. 3B). Unfortunately, due to sample restrictions, we were not able to exchange the ex vivo antigen-specific B cell data with S1-specific B cell enrichment data in the revised manuscript.

Point-to-point-reply Fig 3: (A) Enrichment of S1-specific B cells from PBMC samples of 2 different donors by applying anti-PE microbeads. (B) Progressive development of class switched IgG CD27+ memory cells in these 2 vaccinated individuals.

Discussion: the authors state that the lower CD38 expression level after vaccination (comparing to SARS-CoV-2 infection) might result from limited antigen recognition. This might also be, at least in part, driven by differences in inflammation during infection and vaccination.

We completely agree with this notion by the reviewer and rephrased the respective paragraph in the discussion section accordingly (page 15, lines 337 and 340, of the revised manuscript).

Line 328: “only provide insides’ should read “only provide insights” We have corrected this typo.

Reviewer Reports on the First Revision:

Referee #1 (Remarks to the Author):

The authors addressed satisfactory all my comments.

They performed new experiments to increase the number of vaccinated individuals studied. They also expanded the analysis to a new CD8 T specificity.

The data are robust and fairly discussed. I think the work constitutes a novel and well performed description of the Virus-specific CD8 T cells induced by mRNA vaccination in human over time.

Referee #2 (Remarks to the Author):

The authors have addressed many of the original concerns and have added an additional epitope for analysis. The data continue to be a useful contribution. While they have softened some language, they continue to emphasize the idea that there is a reduced central memory phenotype and “self-renewal” capacity in the vaccinated group. I think this is an incredibly provocative claim that needs to be firmly founded in the data and the statistical analysis. For example, on line 334 they say “these observations my hint towards a restricted self-renewal and maintenance of spike-specific CD8+ T cells after vaccination compared to infection.” Similarly on line 279 they say that vaccination induces a different subset distribution “dominated by effector memory T cells at the expense of more early differentiated subsets with a higher self-renewing capacity and multipotency.” The lack of grounding for these results and repeated use of this language (“at the expense of”) is needlessly provocative and not representative of the actual findings.

1) As quoted above (lines 333-334) they are basing these claims on the BCL-2 and TCF-1 data. The TCF-1 data comparing vaccination and infection is plotted as a t-sne in Figure 4D. This is not statistically comparable and actually these figures are not particularly informative. When I compare by eye the vaccination to the infection, which is all I can do since no summary statistics are provided, I do see that there is a higher number of low TCF1 expressers in the vaccinated group, but there are a higher number of vaccinated “cells” overall. This is not a helpful way to represent these data—we can’t see individual variation, and there are no comparative statistics. The data for BCL-2 are plotted in a much more clear fashion in extended figure 9, but the differences here are not impressive—it is different for 1/3 epitopes at 2/3 timepoints. I suspect the TCF-1 differences are similar. For Tcm it is similarly restricted to 2/3 epitopes at (I think) one time point shown.

2) Related to this, for two of the epitopes, there are many more cells in the vaccinated group than in the infected group—when the phenotyping proportions are reported, can the authors comment on what this means in terms of the actual number of memory cells generated? Even if the vaccine group has a lower proportion in some cases, the higher overall number might compensate.

3) The authors mention that these comparisons are time point matched, but in many graphs (e.g. figure 4) this isn’t clear. What is >80? How far >80 is a comparison allowed? This should be clear throughout.

4) For line 64, the edit is well-taken but I think is still too strong—they can’t claim the CD8 T cells are “important effector cells in early protection after SARS-CoV-2 vaccination” with this kind of study (even if likely!) They could say “are important effector cells expanded in the early protection window” or something like that which makes it clear they are reporting a correlation.

In sum, the data do not support a claim that vaccination generates a phenotype “dominated” (which implies a flip in the majority representation of the cells, which again is only true for one epitope) by effector memory T cells (line 62).

Referee #3 (Remarks to the Author):

The authors addressed all of my concerns.

Author Rebuttals to First Revision:

REFEREES' COMMENTS:

REFEREE #1:

The authors addressed satisfactory all my comments.

They performed new experiments to increase the number of vaccinated individuals studied. They also expanded the analysis to a new CD8 T specificity.

The data are robust and fairly discussed. I think the work constitutes a novel and well performed description of the Virus-specific CD8 T cells induced by mRNA vaccination in human over time.

We would like to thank this reviewer for the positive feedback acknowledging our study.

REFEREE #2:

The authors have addressed many of the original concerns and have added an additional epitope for analysis. The data continue to be a useful contribution. While they have softened some language, they continue to emphasize the idea that there is a reduced central memory phenotype and “self-renewal” capacity in the vaccinated group. I think this is an incredibly provocative claim that needs to be firmly founded in the data and the statistical analysis. For example, on line 334 they say “these observations my hint towards a restricted self-renewal and maintenance of spike-specific CD8+ T cells after vaccination compared to infection.” Similarly on line 279 they say that vaccination induces a different subset distribution “dominated by effector memory T cells at the expense of more early differentiated subsets with a higher self-renewing capacity and multipotency.” The lack of grounding for these results and repeated use of this language (“at the expense of”) is needlessly provocative and not representative of the actual findings.

We would like to apologize since it was not our intention to be provocative. According to this reviewer’s concerns we further softened our conclusions regarding the self-renewing capacity and effector memory subset distribution of spike-specific CD8+ T cells, e.g. we have removed the above-stated conclusions (lines 279 and 334 of the previous version of the manuscript).

1) As quoted above (lines 333-334) they are basing these claims on the BCL-2 and TCF-1 data. The TCF-1 data comparing vaccination and infection is plotted as a t-sne in Figure 4D. This is not statistically comparable and actually these figures are not particularly informative. When I compare by eye the vaccination to the infection, which is all I can do since no summary statistics are provided, I do see that there is a higher number of low TCF1 expressers in the vaccinated group, but there are a higher number of vaccinated “cells” overall. This is not a helpful way to represent these data—we can’t see individual variation, and there are no comparative statistics. The data for BCL-2 are plotted in a much more clear fashion in extended figure 9, but the differences here are not impressive—it is different for 1/3 epitopes at 2/3 timepoints. I suspect the TCF-1 differences are similar. For Tcm it is similarly restricted to 2/3 epitopes at (I think) one time point shown.

Following this reviewer's suggestion, we now additionally depict TCF-1 expression of spike-specific CD8+ T cells in scatter/bar graphs (similar to BCL-2) in Extended Data Fig. 10b. We agree with this reviewer that the described differences in TCF-1 and BCL-2 expression in spike-specific CD8+ T cells after vaccination compared to natural infection show variations and therefore also more precisely describe our results in the Results section (line 234-236).

2) Related to this, for two of the epitopes, there are many more cells in the vaccinated group than in the infected group—when the phenotyping proportions are reported, can the authors comment on what this means in terms of the actual number of memory cells generated? Even if the vaccine group has a lower proportion in some cases, the higher overall number might compensate.

As depicted in Figure 4a, this reviewer is completely right that spike-specific CD8+ T cells targeting A*02/S₂₆₉ and A*03/S₃₇₈ are more frequent after vaccination compared to natural infection. We furthermore agree with this reviewer that these differences in frequencies might compensate for the lower proportions. Based on the fact that the spike-specific CD8+ T cells are still in the dynamic phase of the response 3-4 months after vaccination, we softened our conclusions about the differences in memory subset distribution, emphasize this limitation of our study and the requirement to investigate long-term immunity in follow-up studies in the last paragraph of the Discussion section.

3) The authors mention that these comparisons are time point matched, but in many graphs (e.g. figure 4) this isn't clear. What is >80? How far >80 is a comparison allowed? This should be clear throughout.

We agree with this reviewer that this is an important point. We have precisely described the time-points analyzed in the respective Results section. With respect to Figure 4 the vast majority of samples analyzed were obtained from samples collected d80-120 post boost vaccination or post symptom onset, respectively. Only a few samples were obtained from donors d120-200 post boost vaccination or post symptom onset (line 211-212). As depicted in Extended Data Figure 9, there was no statistically significant difference between the time-points of sample collection after vaccination compared to natural infection.

4) For line 64, the edit is well-taken but I think is still too strong—they can't claim the CD8 T cells are "important effector cells in early protection after SARS-CoV-2 vaccination" with this kind of study (even if likely!) They could say "are important effector cells expanded in the early protection window" or something like that which makes it clear they are reporting a correlation.

We would like to thank this reviewer for this suggestion and rephrased the mentioned sentence accordingly (line 58).

In sum, the data do not support a claim that vaccination generates a phenotype “dominated” (which implies a flip in the majority representation of the cells, which again is only true for one epitope) by effector memory T cells (line 62).

We softened our conclusion, accordingly, removing the statement “dominated by effector memory T cells (line 57).

REFEREE #3:

The authors addressed all of my concerns.

We would like to thank this reviewer.